



# Clouds and precipitation in the initial phase of marine cold air outbreaks as observed by airborne remote sensing

Schirmacher Imke[1], Schnitt Sabrina[1], Klingebiel Marcus[2], Maherndl Nina[2], Kirbus Benjamin[2], Ehrlich André[2], Mech Mario[1], and Crewell Susanne[1]

[1]Institute for Geophysics and Meteorology, University of Cologne, Cologne, Germany
[2]Institute for Meteorology, Leipzig University, Leipzig, Germany

**Correspondence:** Imke Schirmacher (imke.schirmacher@uni-koeln.de)

**Abstract.** Marine cold air outbreaks (MCAOs) strongly affect the Arctic water cycle and, thus, climate through large-scale air mass transformations. The description of air mass transformations is still challenging partly because previous observations do not resolve fine scales, particularly for the initial development of a MCAO, and lack information about cloud microphysical properties. Therefore, we focus on the crucial initial development within the first 200 km over open water of two MCAO events

with different strengths observed during the HALO-(AC)³ campaign. Based on unique sampling of high-resolution airborne remote sensing and in-situ measurements, the development of the boundary layer, formation of clouds, onset of precipitation, and riming are studied. For this purpose, we establish a novel approach, solely based on radar reflectivity measurements, to detect roll circulation that forms cloud streets.

The two MCAO events observed in April 2022 just three days apart occurred under relatively similar thermodynamic con-

ditions. However, for the first event, colder airmasses from the central Arctic led to a marine cold air outbreak index twice that high as for the second event. Thus, the two cases exhibit different properties of clouds, riming, and roll circulations though the width of the roll circulation is similar. For the stronger MCAO, cloud tops are higher, more liquid-topped clouds exist, the liquid layer at cloud top is wider, and the liquid water path, mean radar reflectivity, precipitation rate, and occurrence are increased. These parameters evolve with distance over open water, as seen by, e.g., boundary layer deepening and cloud top height rising.

Generally, cloud streets form after traveling 15 km over open water. After 20 km, this formation enhances cloud cover to just below 100 %. After around 30 km, precipitation forms, though for the weaker event, the development of precipitation is shifted to larger distances. For the stronger event, we detect riming for cloud temperatures below -20°C. The variability of rime mass has the same horizontal scales as the roll circulation implying the importance of roll circulation on precipitation. The detailed observations of the two MCAO events could serve as a valuable reference for future model intercomparison studies.

## 20 1 Introduction

Marine cold air outbreaks (MCAOs) are accompanied by strong air mass transformations. During Arctic MCAOs, cold and dry air flows from the ice-covered central Arctic southward over the open ocean. There, cloud streets form that are well visible in satellite images and transform to cellular convection downstream under extreme surface heat fluxes (Brümmer, 1996).



Especially over open ocean, cloud streets have important implications on the radiative surface energy budget due to their high
albedo induced by liquid cloud tops (Geerts et al., 2022). Moreover, their long lifetimes affect precipitation evolution and
characteristics (Morrison et al., 2012) and thus the Arctic water cycle. Arctic MCAOs can also strongly influence the weather
in the mid-latitudes (Turner and Marshall, 2011).

Over sea ice, dynamical shear instability triggers mesoscale roll convection inside the atmospheric boundary layer (ABL)
that is strengthened by thermal instability over open water (Atkinson and Wu Zhang, 1996). The strong roll circulation, called
secondary flow, forms cloud streets aligned perpendicular to them (Brümmer, 1999). Within updrafts, clouds develop when
moistened air ascends and reaches supersaturation. Within downdrafts, conditions are mostly cloud free as adiabatic heating
causes evaporation of cloud particles. In the Arctic, cloud streets are often mixed-phase clouds (MPCs) that consist of super-
cooled liquid water mostly found at cloud top and precipitating ice particles below. If supersaturation with respect to liquid is
reached, liquid droplets will form and grow by condensation and ice particles will grow by vapor deposition (Morrison et al.,
2012) after they formed from the liquid phase (Ansmann et al., 2008). In MPCs, ice will not grow by vapor deposition at
the expense of liquid since the vapor pressure is mostly higher than the saturation pressure of ice and water (Korolev, 2007).
Relevant ice growth processes in MPCs are aggregation and riming that are important as they determine precipitation. During
riming, supercooled liquid droplets freeze on ice particles, which get larger and denser (Heymsfield, 1982; Erfani and Mitchell,
2017; Seifert et al., 2019). Nevertheless, riming rarely completely depletes liquid because it also reduces the number of ice
particles by precipitation. Riming can be enhanced in updrafts, which lift ice particles and expose them to supercooled liquid
water over a longer period before precipitating (Fitch and Garrett, 2022). Riming has been observed frequently in Arctic MPCs
(McFarquhar et al., 2007; Mioche et al., 2017), even when the liquid water path ($LWP$) is lower than $50\,\mathrm{g\,m^{-2}}$ (Fitch and
Garrett, 2022). Large Eddy Simulations (LES) by Tornow et al. (2021) find riming highly relevant for preconditioning MCAO
cloud regime transitions.

LES identified riming (Tornow et al., 2022), turbulent transport (de Roode et al., 2019), flow divergence, and sharpness
of the marginal sea ice zone (MIZ) (Spensberger and Spengler, 2021; Gryschka et al., 2008) as important factors for the
transformation of the convection regime within cold air outbreaks. Higher initial concentrations of cloud condensation nuclei
delay precipitation formation (Tornow et al., 2021) and higher cloud droplet number concentrations slow down transformation
to open cells, reduce $LWP$, and enhance amount of precipitation (de Roode et al., 2019). A larger amount of ice nucleating
particles (INPs) increases the ice content of clouds, which might enhance riming and thus precipitation (Abel et al., 2017;
Tornow et al., 2021). However, while a sharper MIZ increases surface heat fluxes, it seems not to affect precipitation amount in
contrast to convergences (Spensberger and Spengler, 2021). Furthermore, model settings like the employed ice microphysical
scheme and model resolution affect the timing of transformation that differs between the models (de Roode et al., 2019). Cloud
cover decreases when the ice phase is permitted (de Roode et al., 2019), while a higher resolution evokes roll convection at
smaller fetches and increases precipitation amount (Spensberger and Spengler, 2021). However, no consensus has been reached
between different model studies yet.

A variety of measurements have been performed to gain a better understanding of MCAOs. The theoretical framework about
the secondary flow and geometrical cloud parameters in arctic and tropical (Kuettner, 1971) MCAOs mostly builds on airborne



in situ measurements of the thermodynamic state inside the ABL. Examples for these Arctic campaigns are the Convection and Turbulence (KonTur) experiment (Markson, 1975; Brummer et al., 1982; Brümmer et al., 1985), ARKTIS '88 (Brümmer et al., 1992), '91 and '93 (Brümmer, 1999), and the Marginal Ice Zone Experiment (MIZEX; Walter and Overland, 1984). The influence of the warm ocean on the development of the ABL conditions and cloud/circulation morphology was analyzed by Brümmer (1996) and Müller et al. (1999). They showed that the distance to the sea ice edge characterizes the influence of heat fluxes from open water. When additionally considering open water in the MIZ and over leads, the total distance over open water is called fetch (Spensberger and Spengler, 2021). Variable initial sea ice conditions and convergence of the large-scale flow (Brümmer, 1996) might lead to variable cloud street characteristics across the streets at a specific fetch (Müller et al., 1999).

The transformation of roll to cellular convection has been studied in recent campaigns like the Cold-Air Outbreaks in the Marine Boundary Layer Experiment (COMBLE). During COMBLE, remote sensing and in situ observations measured different stages of cloud development at two ground stations more than 1,000 km away from the sea ice (Geerts et al., 2022). Shipborne measurements, e.g., the Iceland Greenland seas Project (IGP), enable the characterization of convection over time periods longer than research flights (Duscha et al., 2022). The evolution of snowfall rates during MCAOs retrieved from CloudSat Cloud Profiling Radar (CPR) observations were studied by Mateling et al. (2023). However, according to Schirmacher et al. (2023), the accuracy of these CPR precipitation amounts is limited at the surface due to the blind zone, and poor spatial and temporal resolution of the CPR, which is especially important in the initial MCAO phase when the ABL lies inside the blind zone.

Lately, the transformation was studied in a pseudo- and quasi-Lagrangian way by computing back trajectories. Focussing on liquid cloud evolution concerning outbreak strength and aerosol concentration, Murray-Watson et al. (2023) retrieved the time since an air mass passively sensed by a satellite passed the sea ice edge. However, no mixed-phase clouds were considered due to satellite limitations. To characterize the thermodynamic evolution prior to a MCAO event, Dahlke et al. (2022) combined radiosoundings of the same air mass at Ny-Ålesund and RV *Polarstern* during the Multidisciplinary drifting Observatory for the Study of Arctic Climate (MOSAiC) experiment. Not only case studies but also trends of the strength and location of Arctic MCAOs were conducted based on reanalysis data (Papritz and Spengler, 2017; Dahlke et al., 2022). Diminishing sea ice extend, the induced changes in atmospheric dynamics (Dahlke et al., 2022), and clouds (Klingebiel et al., 2023) might lead to strong MCAOs at locations where they did not occur so far.

Despite recent advances in MCAO understanding, a better understanding of microphysical processes is needed to improve numerical simulations of past, present, and future MCAOs. In particular, it is important to disentangle the roles of local and large-scale influences on roll circulation evolution. However, the difficulty in resolving the initial MPC conditions remains when using satellite observations that cover large distances. Spatially fine resolved observations across multiple cloud streets, particularly close to sea ice edge, exist only rarely for cloud macrophysical properties and, to our knowledge, not at all for cloud microphysical properties and precipitation.

This study analyzes high-resolution airborne remote sensing and collocated in situ observations of MCAOs close to the sea ice, where conditions strongly shape further downstream MCAO development. Dedicated flight patterns sampled repetitive





crosssections through cloud streets on two days during the HALO-(AC)[3] (High Altitude and LOng range research aircraft

- ArctiC Amplification: Climate Relevant Atmospheric and SurfaCe Processes, and Feedback Mechanisms; Wendisch et al., 2021) campaign in March/April 2022. Trajectory calculations allow us to analyze ABL and cloud development, precipitation onset, and riming in a pseudo-Lagrangian way, i.e., as a function of fetch. The focus is on the initial development within the first 170 km fetch. We aim to resolve the variability of meteorological parameters that is unresolved in grid boxes of reanalyses and numerical weather predictions, and even in satellite observations like the $LWP$ from the Moderate Resolution Imaging

Spectroradiometer (MODIS; Fig. 1a, b; red box in (b) compared to (d)). The area of interest is the Fram Strait since most strong MCAOs in the Nordic Seas occur west of Svalbard (Papritz and Spengler, 2017). Contrarily to previous flight strategies, our flight paths cross the cloud streets perpendicularly and, thus, go along the secondary flow to enable its investigation. To investigate the phase composite of MPCs, we use remote sensing observations taken from above clouds, which resolve the liquid layer at cloud top best. With an averaged rime fraction of 87 %, Maherndl et al. (2023b) showed that riming is a key

process during the full HALO-(AC)[3] campaign when remote sensing and collocated in situ observations were obtained. Thus, we aim to answer the open question of how riming depends on fetch and how it relates to the roll convection for the two case studies.

The paper is organized as follows: First, we introduce the airborne instruments and data, as well as auxiliary spaceborne and reanalysis data (Sect. 2). Second, we describe our methodology: the calculation of fetch for observations from back trajectories

(Sect. 3.1) and the identification of roll circulations from radar reflectivities (Sect. 3.2). Section 4 answers the following questions:

   I.   Which thermodynamic conditions characterize the two MCAO events (Sec. 4.1)?

   II.  Which cloud properties are associated with roll circulation (Sec.4.2)?

   III. How do circulation and cloud properties change with fetch in the initial state of MCAOs and when do cloud streets start

115       to precipitate (Sec. 4.3)?

   IV.  What is the impact of riming on MCAO transformation? (Sec. 5)

The findings are synthesized to describe the interaction between roll circulation, cloud macro-, and microphysics (Sect. 6). Finally, Sect. 7 concludes the study and discusses future steps.

## 2   Data

In this study, airborne measurements from the HALO-(AC)[3] campaign (Wendisch et al., 2021) are analyzed. During this campaign, the High Altitude and LOng-range research aircraft (*HALO*; Ziereis and Gläßer, 2006), the research aircraft *Polar 5* (*P5*) and *Polar 6* (*P6*; Wesche et al., 2016) operated in the North Atlantic sector of the Arctic at altitudes around 10 km, 3 km and below 3 km, respectively. This analysis mostly focuses on radar, radiometer, lidar, and dropsonde measurements from *P5* that probed MCAO events in their early phase. Dropsonde measurements from *HALO* and in situ observations from *P6* further

support the analysis. We limit the analysis to measurements taken over ocean and restrict the remote sensing measurements to straight flight segments that exceed a flight altitude of 2 km observing clouds from aloft. The focus lies on two MCAO events,

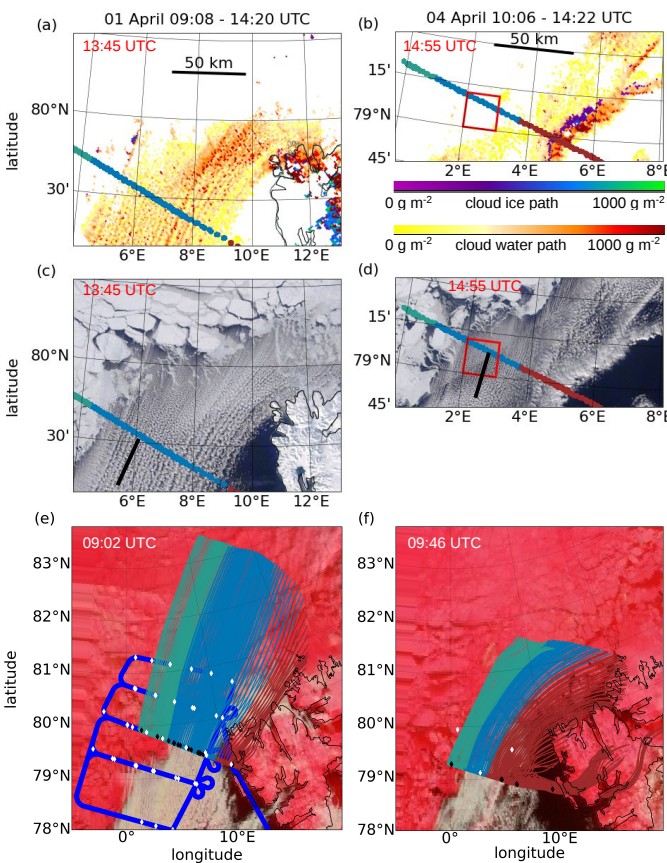

**Figure 1.** Cloud situation on 01 April 2022 (left column) and 04 April 2022 (right column). Observations of cloud water and ice path (Nasa Worldview, 2023a; a, b; 1 km resolution) and corrected reflectance (Nasa Worldview, 2023b; c, d; 500 m resolution) of the Moderate Resolution Imaging Spectroradiometer (MODIS) on Terra on 01 April (13:45 UTC) and 04 April (14:15 UTC). The black lines in (c) and (d) represent the orientation of the cloud streets. To better differentiate between sea ice and clouds, the corrected reflectance observed by the Visible Infrared Imaging Radiometer Suite (VIIRS) instrument aboard the joint NASA/NOAA Suomi National Polar-orbiting Partnership (Nasa Worldview, 2023c; e, f) at 09:02 UTC on 01 April and 09:46 UTC on 04 April is shown. Additionally, 12-hour near-surface back trajectories for the measurement locations (thin colored lines), dropsonde locations (diamonds) of *P5* (black) and *HALO* (white), and the flight path of *HALO* on 01 April 2022 (blue line) are shown. The flight path of *P5* (thick colored line) overlays all maps. Here, the colors categorize the measurement regimes (Sect. 3.1, Table 1).

namely on 01 April 2022 (Fig. 1c) and on 04 April 2022 (Fig. 1d), during which cold air was advected from the sea ice over the Fram Strait leading to the formation of cloud streets. On 04 April, a convergence line also appeared in the Fram Strait's center. Since we aim to investigate the influence of varying meteorological conditions on roll circulation, we analyze both case studies, which show slightly varying environmental conditions. To investigate the roll circulation, the flight paths crossed the






**Table 1.** Categorization of *P5* airborne data.

| day | description | color | location |
|---|---|---|---|
| 01 April | influence by Svalbard | red | longitude>9.08°E |
| 01 April | prior to cloud streets | green | fetch<15 km |
| 01 April | cloud streets | blue | remaining data |
| 04 April | influence by Svalbard and convergence line | red | longitude>3.7°E |
| 04 April | prior to cloud streets | green | longitude<1.7°E |
| 04 April | cloud streets | blue | remaining data |

cloud streets perpendicularly. *P5* probed along the same path back and forth, yielding 6 legs on 01 April (09:08–14:20 UTC) and 4 legs on 04 April (10:06–14:22 UTC).

On both days, *P5* and *P6* were closely collocated. For the analysis of the collocated flights, we use a data subset during which both aircraft flew on straight paths with a time difference between the collocated measurements of less than 5 min, a

spatial distance between both platforms below 5 km, and a flight altitude of *P6* between 0.15 and 1.3 km. With this, we reduce the error caused by sampling different air masses with *P5* and *P6* and also by sampling air masses with varying microphysical properties due to changing *P6* locations within the cloud vertical extent. On 01 April, 3971 seconds of collocated observations cover longitudes between 4.5 and 6.5°E corresponding to 25–165 km fetch. On 04 April, only 845 seconds of observations are collocated that are located between 1.5 and 4.5°E and cover fetches between 55 and 165 km, mostly at around 80 km. On 01

April, seven collocated data segments exist with gaps of less than 5 s. These segments cover 39 min at 60–140 km fetch with most measurements concentrated around 7°E longitude.

## 2.1 Instruments

**Dropsondes:** Vaisala Dropsondes RD94 were launched from *P5* and *HALO*. From *P5*, 18 and 14 sondes were launched on 01 and 04 April, respectively. They provide vertical profiles of potential temperature ($\theta$; accuracy=0.2 K), relative humidity (2

%), pressure (0.4 hPa), and horizontal wind components derived from GPS recordings (Vaisala, 2010; George et al., 2021).

**Microwave Radar/radiometer for Arctic Clouds (MiRAC):** The active component of the downlooking airborne MiRAC (Mech et al., 2019) onboard *P5* consists of a frequency-modulated continuous wave (FMCW) radar that operates at 94 GHz. Additionally, a 89 GHz passive channel accompanies the active measurements. MiRAC measures every second, which corresponds to a horizontal resolution of the equivalent radar reflectivity ($Ze$) of about 85 m at ground in flight direction for a typical

cruise altitude of 3 km height and ground speed of 80 m s$^{-1}$. The radar measurements are quality controlled and corrected for the 25° backward inclination of the instrument, surface clutter, and aircraft attitude (Mech et al., 2019). The sensitivity and vertical resolution of the cloud radar depend on the setting. During HALO-(AC)$^3$, the detection limit for the most distant ranges of 3 km from *P5* was around -45 dBZ and the vertical resolution was 4.5 m close to the aircraft and at most 13.5 m (Mech et al., 2022a). The processing interpolated the vertical resolution to 5 m over the whole profile. A blind zone of 150 m above ground

is omitted due to ground clutter (Schirmacher et al., 2023). The accuracy of $Ze$ is about 0.5 dBZ. Attenuation by water vapor



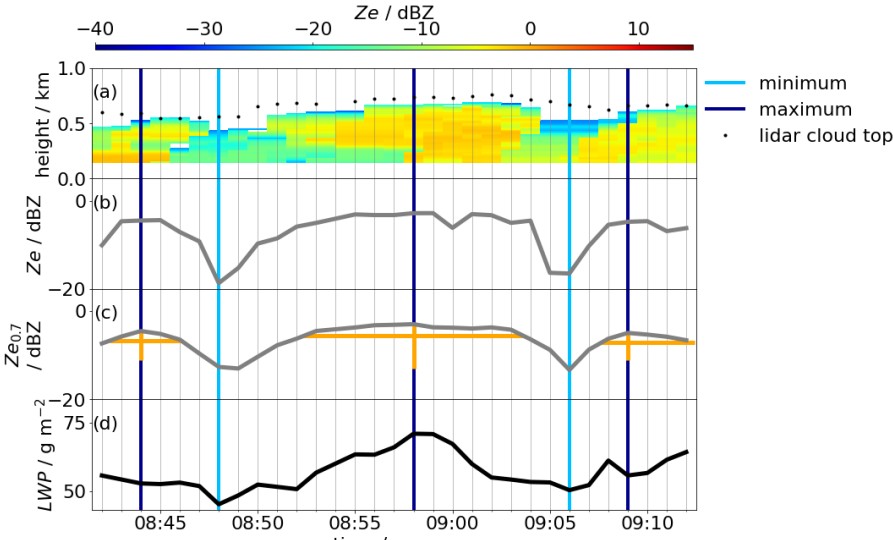

**Figure 2.** Visualization of cloud circulation identification. Time series (30 s) of the equivalent radar reflectivity $Ze$ profiles measured by MiRAC starting at 10:08:43 UTC on 01 April (a), $Ze$ at an altitude of 0.7 of the normalized hydrometeor depth (b, step III), and after further averaging over 3 s (c, step V). The detection of maxima (dark blue, step VI) and minima (light blue, step VII) is described in Sec. 3.2. The prominence (c, vertical orange line) and width of the peak (c, horizontal orange line) are important parameters for the peak detection. Note that $Ze$ is shown in logarithmic space while peaks are detected in the linear space. Time series of liquid water path ($LWP$, 25° back inclination) shifted by 16 s to match nadir-looking $Ze$ observations for clouds with a cloud top height of 1 km (d). For comparison, the cloud top height observed by the lidar AMALi is displayed (a, black dots). The shown time period covers fetches from 73 to 80 km and corresponds to a flight distance of 7 km.

(<1 dBZ) and clouds (∼0.6 dBZ) can potentially reduce this accuracy (Schirmacher et al., 2023). For some profiles, the radar detects several cloud layers. However, we focus only on the highest cloud layer and do not differentiate between multilayer clouds in the analysis. The passive channel observes brightness temperatures ($TB$) primarily influenced by emission of liquid clouds and the surface. Differences in $TB$ for clear-sky and cloudy situations are used to retrieve $LWP$ over ocean via a

regression approach (Ruiz-Donoso et al., 2020). Its sensitivity is below $5 \, \mathrm{g \, m^{-2}}$ and the absolute accuracy below $30 \, \mathrm{g \, m^{-2}}$ (Ruiz-Donoso et al., 2020). While radar reflectivities are corrected to nadir profiles, the $LWP$ is derived from slanted $TB$ measurements (Mech et al., 2022a). Because of the liquid layer at cloud top, the $LWP$ signal of MPCs is mostly from the uppermost few hundred meters of the clouds. Due to variable cloud top heights ($CTHs$), $LWP$ lags behind the radar observation in a non-constant matter. However, based on geometric considerations, we shift the $LWP$ measurements assuming a daily

average $CTH$ for cloud streets. Since this average differs for both days, we shift the $LWP$ measurements by different time periods, i.e., 16 and 19 s on 01 and 04 April, respectively, having an estimated maximum error of 4 s. As a result, $LWP$ peaks coincide with profiles of high $Ze$ (Fig. 2a, d).



**Airborne Mobile Aerosol Lidar (AMALi):** AMALi onboard *P5* measures profiles of backscattered intensities at 532 nm (parallel and perpendicular polarized) and 355 nm (not polarized; Stachlewska et al., 2010). With a vertical resolution of 7.5 m, the $CTH$ is obtained for every profile that has consecutive heights with backscatter coefficients exceeding the one of cloud-free sections by a factor of five. The $CTH$ is the maximum altitude of these consecutive heights. Further details can be found in Mech et al. (2022a) and Schirmacher et al. (2023).

**In situ probes:** The *P6* was equipped with three in situ probes, namely the Cloud Droplet Probe (CDP; Lance et al., 2010), Cloud Imaging Probe (CIP; Baumgardner et al., 2011) and Precipitation Imaging Probe (PIP; Baumgardner et al., 2011). The CDP is a forward-scattering optical spectrometer that measures small cloud particles (2.8–50 $\mu$m). Larger cloud particles are observed by the CIP (15–960 $\mu$m) and PIP (103 $\mu$m–6.4 mm) that record shadow images of the cloud particles as the particles pass through the sampling area (Moser et al., 2023). By combining CDP, CIP, and PIP, a continuous particle size distribution is derived for calculating rime mass. The CIP and PIP data are processed similarly to previous campaigns (Mech et al., 2022a).

## 2.2 Satellite and reanalysis data

For the sea ice concentration (SIC), we use a daily product that merges satellite observations from MODIS and the second Advanced Microwave Scanning Radiometer (AMSR2) at 1 km horizontal resolution (Ludwig et al., 2020). For the analysis, we interpolate the data to a 0.05° x 0.05° latitude/longitude grid. Sea surface temperatures ($SST$) are obtained from the Arctic Ocean - Sea and Ice Surface Temperature product based upon observations from the Metop–A Advanced Very High Resolution Radiometer (AVHRR). The daily product (Copernicus Marine Service, 2023) has a spatial resolution of 0.05° and covers surface temperatures of the ocean, sea ice, and MIZ. Using satellite $SST$ and dropsonde temperature measurements above open water (Fig. 1e, f), we calculate the $MCAO$ index from the difference between the potential temperature ($\theta$) at sea surface and 850 hPa altitude. Generally, the $MCAO$ index is positive during a MCAO and describes its strength (Papritz et al., 2015; Kolstad, 2017). To compute back trajectories using Lagranto (Sprenger and Wernli, 2015), we use the three-dimensional wind fields of the European Centre for Medium-Range Weather Forecasts (ECMWF) reanalysis product version 5 (ERA5; Hersbach et al., 2020). This product's temporal, horizontal, and vertical resolution is one hour, 31 km, and 137 model levels from the surface up to the top of atmosphere, respectively (Kirbus et al., 2023).

## 3 Analysis and identification approaches

In the following, we describe two approaches we established for analyzing airborne measurements. The first concerns the quantification of the fetch assigned to each measurement (Sect. 3.1). Second, we present a novel method to identify roll circulation from airborne radar reflectivities only (Sect. 3.2).

### 3.1 Trajectory calculations and fetch

The warm ocean alters the ABL conditions by turbulent surface heat and moisture fluxes (e.g., Brümmer, 1996). The thermodynamic profile of an airmass formed in the central Arctic flowing southward is modified by strong turbulent surface fluxes





evident whenever SIC is below 100 % even if no clouds develop. We aim to quantify this influence of open water on the ABL
development. The distance of an air parcel to sea ice edge has been used to characterize the exposure to open water (Brümmer,
1996; Müller et al., 1999). However, this distance does not capture the influence of MIZ and leads encountered in the ice. We
account for all areas of open water, which was already done in the past by either calculating the fetch (Brümmer, 1996; Müller
et al., 1999; Spensberger and Spengler, 2021) or travel time (Tornow et al., 2021; Murray-Watson et al., 2023).

Turbulent fluxes are proportional to wind speed. Stronger fluxes supply more heat/moisture, affect higher altitudes, and,
thus, modify the atmosphere more. If winds of different strengths advect an air mass over a constant fetch, the integrated flux
will be roughly constant for both scenarios since the shorter travel time counteracts the stronger fluxes for the higher wind
speed. If winds of different strengths advect an air mass for a constant time, the instantaneous fluxes, integrated flux, and fetch
increase for the higher wind speed. Therefore, the integrated flux depends on wind speed similar to fetch as opposed to travel
time. Hence, we calculate fetch for each airborne measurement to quantify the influence of open water.

In this study, however, the flow is slowed to an extent that discrepancies between travel time and fetch emerge only over
land. The correlation coefficient between the travel time over open water and fetch is 0.99 for all *P5* measurements that are not
influenced by land masses and -0.5 for *P5* measurements that are influenced by Svalbard. For wind flows unaffected by land
masses, integrated travel time and fetch should be linearly convertible and both parameters are valid to study.

To calculate fetch, we need to know the air masses' previous path. For this purpose, we compute near-surface Lagrangian
back trajectories using Lagranto (Sprenger and Wernli, 2015) with ERA5 wind fields as input. ERA5 captures ABL conditions
well, e.g., it represents the course of the ABL height evolution along the flight path similarly to radar observations even though
it overestimates the absolute values by roughly 200 m (not shown). Specifically, we calculate back trajectories for the previous
12 hours for every flight minute and resample them to one second. The trajectories originate at the horizontal location of *P5*
and at 1000 hPa height corresponding to roughly 300 m above the surface. To investigate the influence of the surface on the air
masses, we take a near-surface starting point for the trajectories. Similar to Spensberger and Spengler (2021), we calculate the
fetch for every back trajectory from the ERA5 wind field by integrating the ratio of open water obtained from MODIS-AMSR2
SIC data (Sect. 2.2) along the back trajectory paths over the previous 12 h until measurement time (0 h; see Appendix A for
error estimation):

$$fetch = \int\limits_{s(12\,h)}^{s(0\,h)} (1 - SIC(s))ds \tag{1}$$

To concentrate on cloud street characteristics during undisturbed MCAO conditions, we limit the analysis to data that did
not pass Svalbard at any time and that are not affected by the convergence line on 04 April (Tab. 1; Fig. 1, non-red dots). The
remaining measurements are classified to be either 'cloud streets' (Fig. 1, blue) if radar reflectivities appear regularly or 'prior
to cloud streets' (Fig. 1, green; Tab. 1). The latter category includes samples taken over and close to sea ice that have fetches
less than 15 km on 01 April and longitudes smaller than 1.7°E on 04 April (fetches of about 17 km). On 04 April, longitude
instead of fetch is used for classification because cloud streets over open ocean and cloud-free conditions over sea ice have both
fetches of 17 km. Note that fetch includes contributions from the MIZ (80 % <SIC> 100 %) and open water. Due to the low





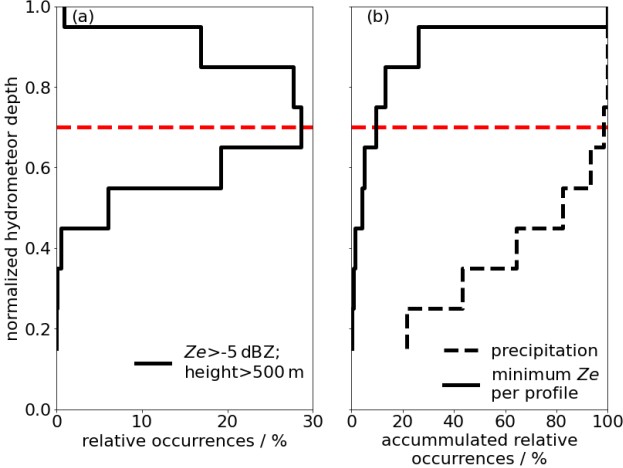

**Figure 3.** Relative occurrences of $Ze$ that exceed -5 dBZ and are above 500 m height (a). Relative occurrence of precipitation (b, dashed) and minimum $Ze$ of each radar profile (b, solid) as a proxy for dry entrainment accumulated with height. The y-axis is the normalized hydrometeor depth (0=base, 1=top). The height that is used to identify roll circulations is indicated by the red line.

amount of data that have variable fetches due to different fetches over MIZ only, it is impossible to disentangle the influence of MIZ.

### 3.2 Roll circulation identification

235 To identify roll circulations from radar measurements, we must rely on indirect information. Previous studies applied spectral analysis to observations of the three wind components, temperature, mixing ratio, and radiative fluxes (Brümmer et al., 1985, 1992; Brümmer, 1999; Walter and Overland, 1984). While vertical velocity cannot be extracted from the airborne Doppler measurements (Mech et al., 2022a), we exploit the fact that due to the vertical motion, cloud particles form to the largest extent at the location of the strongest updraft if they are not influenced by dry entrainment at cloud top (Klingebiel

240 et al., 2015) or precipitation (Morrison et al., 2012). This leads to a maximum in the radar signal. We solely use $Ze_{0.7}$, which is the radar reflectivity measurements at the height of 0.7 of the hydrometeor depth ($D$, the difference between $CTH$ and lowest radar signal height per profile). In this way, $Ze_{0.7}$ serves as a proxy for vertical velocity, and we assume that maxima in $Ze_{0.7}$ represent updraft and minima downdraft regions of the roll circulation.

We take $Ze$ at the height of 0.7 of $D$ to minimize the influence of dry air entrainment/supercooled liquid water droplets at

245 cloud top and precipitation at the bottom of $D$. Precipitation is assumed to occur at $Ze$ values larger than -5 dBZ (Schirmacher et al., 2023) that are below 500 m height (Shupe et al., 2008). Figure 3 explains the choice of this height (red line): here, non-precipitating large ice particles ($Ze > $ -5 dBZ, height $> 500$ m) are most frequent with 29 % (Fig. 3a) and with 98 %, most precipitation occurs below that height level (Fig. 3b, dashed). Dry entrainment and liquid droplets seem rare at that height since





91 % of the lowest $Ze$ per profile lie above the altitude of 0.7 of $D$ (Fig. 3b, solid). In conclusion, we take $Ze$ as a proxy for
vertical motion at a height mostly consisting of large ice particles and is least affected by entrainment and precipitation.

To find up- and downdraft regions, we follow this recipe:

  I. Determine the hydrometeor depth ($D$) for every profile.

  II. Average $Ze$ over 100 m in the vertical to reduce noise.

  III. Smooth $Ze$ by averaging over 3 s to minimize noise detection.

IV. Retrieve the smoothed $Ze_{0.7}$ at 0.7 of $D$ for each profile (Fig. 2c).

  V. Derive the large-scale background $Ze_{\mathrm{back}}$ by averaging $Ze_{0.7}$ over 500 s ($\sim 40$ km).

  VI. Determine peaks in $Ze_{0.7}$ using the python package scipy.signal.find_peaks (Virtanen et al., 2020).

  If $Ze_{\mathrm{back}} \geq 0.67 \, \mathrm{mm}^6 \, \mathrm{m}^{-3}$, find peaks for $Ze_{0.7}$ (Fig. 2, dark blue) with a prominence of at least $0.5 \, \mathrm{mm}^6 \, \mathrm{m}^{-3}$ (differ-
  ence between the height of the peak and its lowest contour line; vertical orange line) and a width of at least 2.9 samples
(horizontal orange line).

  If $Ze_{\mathrm{back}} < 0.67 \, \mathrm{mm}^6 \, \mathrm{m}^{-3}$, find peaks for $Ze_{0.7}$ with a prominence of at least $0.1 \, \mathrm{mm}^6 \, \mathrm{m}^{-3}$ and a width of at least 2.9
  samples.

  Here, we apply two different thresholds depending on $Ze_{\mathrm{back}}$ since the magnitude of the averaged $Ze$ and its peaks
  generally increase with fetch.

VII. Find the minimum $Ze_{0.7}$, which is not averaged over 3 s, between every two maxima (Fig. 2, light blue). If conditions
  between two cloud streets are cloud free, we consider the downdraft location at the center of the cloud-free distance.

Note that we can only apply the algorithm to the 'cloud street' regime as roll convection is mostly too weak to evoke radar
signals prior. According to our definition, the maximum updraft, i.e. maximum $Ze_{0.7}$, does not necessarily need to be centered
between the two detected edges of our roll circulation object. The wavelength of the circulation ($\lambda$) is the distance between two
identified adjacent downdrafts. The mesoscale circulation is described by the aspect ratio ($AR$), i.e., the ratio between $\lambda$ and
$CTH$ at the position of the identified maximum in $Ze_{0.7}$. We identified 364 and 109 cloud circulation objects in the 'cloud
street' regime on 01 and 04 April, respectively.

We tested several configurations of the roll circulation detection algorithm (for details, see Table A1) and selected the one
with the best ratio between determining peaks and ignoring noise. This automated peak detection depends only on the large-
scale condition and thus might not determine every maximum of $Ze_{0.7}$ considered by human eye.

## 4   Variability of thermodynamic conditions and cloud street properties

In the following, we first investigate the thermodynamic state of the ABL (Sect. 4.1) to characterize the environmental condi-
tions. For the 'cloud street' regime, we analyze cloud morphological and microphysical properties within the roll circulations
(Sec. 4.2). Afterward, we investigate how roll circulations and thus cloud properties develop as a function of fetch within the
first 170 km of the MCAO development (Sec.4.3 ).



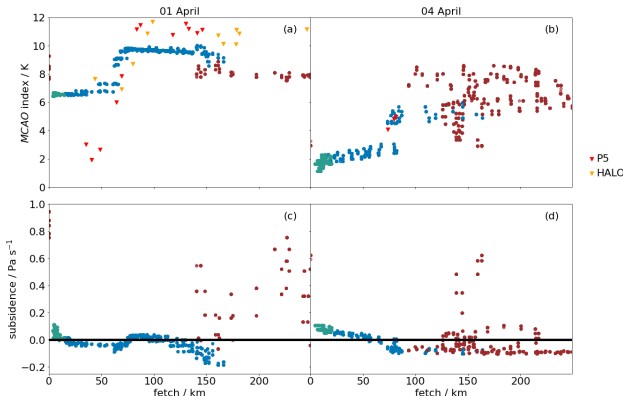

**Figure 4.** Evolution of $MCAO$ index (first row) and subsidence at 1000 hPa (second row) with fetch on 01 April (a, c) and 04 April (b. d) from ERA5 reanalysis data. Positive subsidence values indicate downward motion. The colors categorize the measurement regimes (Sect. 3, Table 1). The triangles in (a) and (b) show the $MCAO$ index retrieved from dropsonde measurements of the research aircraft *P5* (red) and *HALO* (orange).

For the microphysical analysis, we introduce two metrics: first, the supercooled liquid layer thickness ($LLT$), which is the difference in $CTH$ measured by the lidar and radar. Here, we exploit the fact that the lidar is more sensitive to particle amount (liquid), whereas the radar is more sensitive to particle size (ice; Ruiz-Donoso et al., 2020). Due to limited vertical resolutions of the instruments (Sec. 2.1) and resulting uncertainties in $CTH$, the $CTH$ of the lidar has to exceed the $CTH$ of the radar by

at least 10 m to be defined as liquid topped. Second, we define profiles containing a $Ze$ value higher than -5 dBZ (Schirmacher et al., 2023) in the lowest 500 m (Shupe et al., 2008) as precipitating. Using the $Ze$-$S$ relation for three bullet rosettes following Kulie and Bennartz (2009), this value corresponds to a snowfall rate ($S$) of 0.07 mm h$^{-1}$. Note that this is a rough estimate since in situ measurements show that the crystal shapes within cloud streets are very variable (Maherndl et al., 2023a; Moser et al., 2023). In this study, $S$ is analyzed close to the ground at 150 m. The precipitation fraction denotes the ratio of the number

of precipitating to all profiles.

## 4.1 Thermodynamic evolution and cloud appearance

On 01 and 04 April, cold air was advected from the central Arctic over the Fram Strait. With an $MCAO$ index averaged over all dropsondes of 8.6 K, the first MCAO event on 01 April is stronger than the second one on 04 April (Table 2). On both days, the evolution of the $MCAO$ index obtained by ERA5 reanalysis follows the one of $SST$ and the largest increase of the

$MCAO$ index is at around 70 km fetch (Fig. 4a, b), which is twice the size of an ERA5 grid point.

On 01 April, the MODIS image shows cloud streets with an orientation of about 10° to north (Fig. 1c, black line) and a wavelength of about 2 km with shorter distances between the separated streets close to sea ice. Note that the coarsest spatial resolution of the used MODIS sensors (bands 1, 3, and 4) is 500 m. On 04 April, the MODIS image shows cloud streets with an orientation of 5° to north (Fig. 1d, black line) and a wavelength of about 1 km. Due to its topography, the off-land flow at





**Table 2.** Conditions during 01 April and 04 April. $BLH$ stands for atmospheric boundary layer height.

| parameter | source | 01 April | 04 April |
|---|---|---|---|
| $MCAO$ index | dropsondes | 8.6 K | 4.6 K |
| cloud street orientation | MODIS | 10°N | 5°N |
| cloud street wavelength | MODIS | 2 km | 1 km |
| temperature at cloud top | dropsondes | <-20°C | -20 to -10°C |
| $BLH$ trend | dropsondes | 4.5 m km$^{-1}$ | 2.9 m km$^{-1}$ |
| mixing ratio trend within 100 km fetch | dropsondes | < doubling | > doubling |
| driver of wind shear | dropsondes | wind direction | wind speed |
| precipitating radar profiles | radar | 67 % | 35 % |
| liquid-topped radar profiles | radar and lidar | 86 % | 71 % |

1000 hPa westward of Svalbard descends in the vicinity of the island, particularly on 01 April (Fig. 4c, d; red), and the lee effect affects air mass characteristics, e.g., causing cloud-free conditions close to the island. On 01 April, the near-surface air mass of the 'prior to cloud streets' regime subsides (Fig. 4c; green) and of the 'cloud street' regime ascends except for fetches between 75 and 120 km (about 7°E longitude; blue) even though $SST$ and $MCAO$ indices abruptly increase (Fig. 4a; blue). Here, air mass subsides over the whole atmospheric column (not shown) thus we suggest that this subsidence is due to a wave

effect provoked by the island. On 04 April, the air mass starts ascending at larger fetches compared to 01 April without a notable wave effect (Fig. 4d).

The thermodynamic state of the ABL is described by mean profiles of dropsondes released from *P5* and *HALO* over sea ice and open water during both days (Fig. 1e, f, diamonds). Generally, the temperatures within the lowest 1 km increase with fetch due to heating from below (Fig. 5a, f). The temperature profiles reveal the difference in air mass between both days. On 01

April, temperatures are lower than -20°C throughout all altitudes over sea ice (fetch <15 km) and for parts over open water. On 04 April, contrarily, all temperature profiles below 2 km height lie within $-20$ to $-10$°C.

The potential temperature ($\theta$) of the free troposphere is on average lower by about 5 K on 01 April compared to 04 April (Fig. 5b, g). Over open water, a neutrally-stratified ABL develops due to strong sensible and latent heat fluxes from the ocean and turbulent mixing of near-surface air. Note, that ERA5 shows about twice as high fluxes on 01 April compared to 04 April.

Generally, the boundary layer height ($BLH$), i.e., the inversion height of $\theta$, increases with fetch. On 01 April, the averaged $BLH$ increase rate over all fetches is much stronger than on 04 April (Table 2). Over sea ice, the surface layers that are cooled from the ground and the air above that is warmed by subsidence generally develop an inversion. Profiles that were sampled by *HALO* dropsondes over sea ice close to the ice edge on 01 April exhibit a thin ($< 250$ m deep) ABL, while dropsondes cannot detect a $BLH$ in the central Arctic (81.3–87.0°N) on 04 April as it is likely too shallow (not shown). The flow over

more frequent areas of open sea ice or refrozen leads within the MIZ (Li et al., 2020) might deepen the ABL. The inversion is stronger over sea ice and weakens less with fetch on 04 April due to a layer of warm air above the $BLH$. This layer additionally evokes a second inversion over water (Fig. 5g, blue).





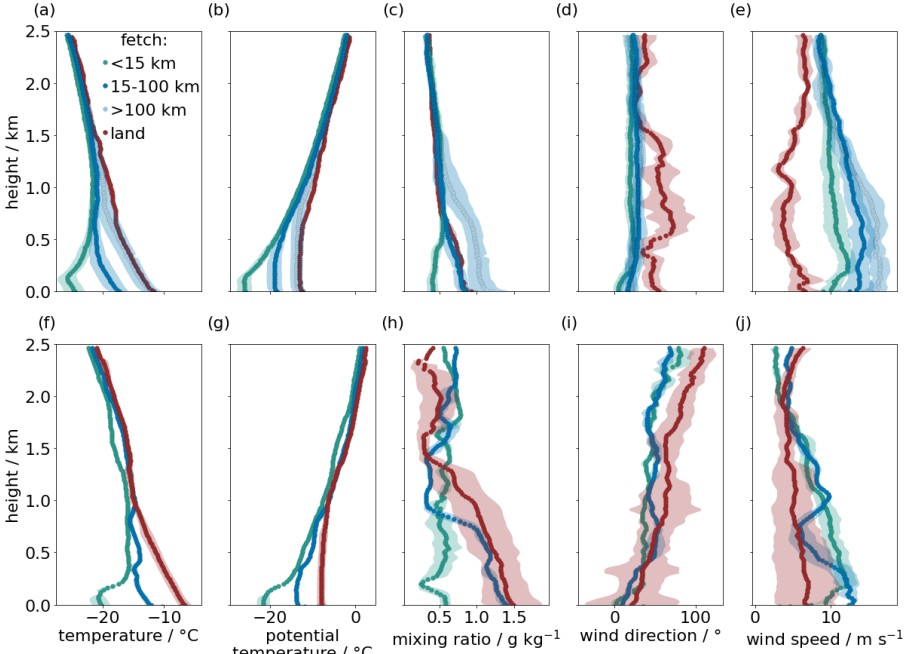

**Figure 5.** Averaged dropsonde profiles from *HALO* and *P5* of temperature (a, f), potential temperature (b, g), mixing ratio (c, h), wind direction (d, i) and speed (e, j) binned by fetch on 01 April (first row) and 04 April 2022 (second row). The shaded areas represent the standard deviation of each category. The color coding is inspired by the categorization shown in Table 1. On 01 April, the number of dropsondes per category is 27 (<15 km fetch; green), 14 (15-100 km and >100 km fetch; dark and light blue), and 3 (land; red). On 04 April, the number is 3, 4, and 9, respectively.

Over sea ice (fetch < 15 km), the low water vapor mixing ratio ($0.5 \, \mathrm{g \, kg^{-1}}$) indicates the background concentration of the polar air mass on 01 April, which is slightly higher over all heights on 04 April. In the ABL, the mixing ratio increases with

fetch due to latent heat fluxes leading to slightly less and more than a doubling within the first 100 km on 01 and 04 April, respectively.

With 28°, wind direction is constant with height on 01 April (Fig. 5d). On 04 April, a shear from north wind at the surface to west wind exists over all heights, which is strongest at 0.4–1 km (Fig. 5i). The near-surface wind on 01 April and boundary-layer wind on 04 April are more northerly, particularly over sea ice they reach 0° N. On both days, the wind speed increases

towards surface before it declines towards the surface at around 200 m (Fig. 5e, j). On 01 April, wind speed increases with fetch, while on 04 April, it reduces by $5 \, \mathrm{m \, s^{-1}}$ over water at 0.4–1 km height, which aligns with the airmass identification from the $\theta$ profiles. The wind inside the ABL follows an Ekman spiral caused by friction that is enhanced over sea ice. Generally, inflection points in the wind form roll circulation by shear instability 14° to the left of the geostrophic wind (Atkinson and Wu Zhang, 1996). On 01 and 04 April, the cloud streets align 10 and 70° to the left of the geostrophic wind, respectively, due



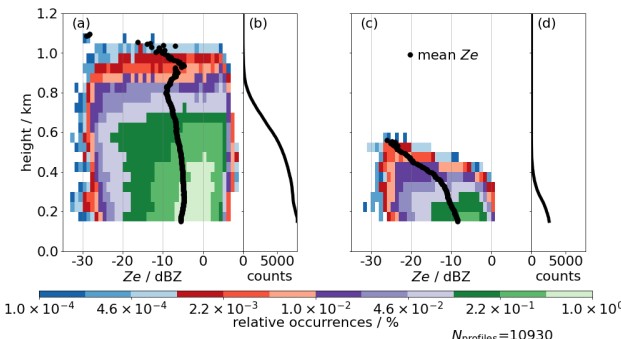

**Figure 6.** Contoured frequency by altitude diagram (a, c) and absolute counts per altitude (b, d) for all radar reflectivities ($Ze$) obtained by MiRAC in the 'cloud street' regime on 01 April (a, b) and 04 April (c, d). Moreover, each averaged $Ze$ profile (black dots), and the total number of profiles ($N_{\mathrm{profiles}}$) is displayed.

to additional convective instability. The observed cloud streets form along the wind direction averaged over the lowest 100 m as found by Puhakka and Saarikivi (1986).

On both days, the radar profiles in the 'cloud street' regime frequently (93 %) exhibit clouds. The contoured frequency by altitude diagrams (Fig. 6) reveal the different cloud and precipitation characteristics of cloud streets between the days: especially, $CTH$ is twice as high on 01 than on 04 April as indicated by the radar measurements. Furthermore, the mean $Ze$

profile (black dots) is larger over all heights. On 01 April, values larger than -5 dBZ, which are associated with the onset of snowfall, occur over all heights, while on 04 April even below 500 m, $Ze$ rarely exceeds -5 dBZ. Thus, more profiles precipitate on 01 than on 04 April (Table 2). On 01 April, mean $Ze$ slightly decreases close to the surface even stronger for small fetches (not shown). Thus, near-surface ice particles might experience stronger evaporation on 01 April, especially when the mixing ratio is comparably small. Hence, surface snowfall rates are overestimated if derived from higher altitudes (Schirmacher et al.,

2023). On 04 April, mean $Ze$ increases towards the surface indicating the ongoing growth process of ice particles. Most cloud streets are liquid-topped (Table 2). For most of these clouds, the $CTH$ is higher than for non-liquid-topped clouds, and the averaged $Ze$ profile is larger over all heights on 01 April (Fig. A1). Stronger turbulence in liquid-topped clouds might lift particles higher and increase amount and size of ice particles (Morrison et al., 2012).

In summary, the MCAO event is colder and drier on 01 than on 04 April with higher $MCAO$ indices (Table 2). On both

days, wind shear occurs by wind speed changes on 01 and direction changes on 04 April. On 04 April, strong changes in wind direction at 0.4–1 km over open water result in a slower advection of warm and moist air from the direction of Svalbard. Thus, the strength of the temperature inversion above $BLH$ decreases less with fetch over open water than on 01 April, which reduces the rate of $BLH$ increase with fetch (Brümmer, 1996). Contradicting the $MCAO$ index, ERA5 indicates subsidence at the surface in the 'cloud street' regime at fetches around 100 km on 01 April.



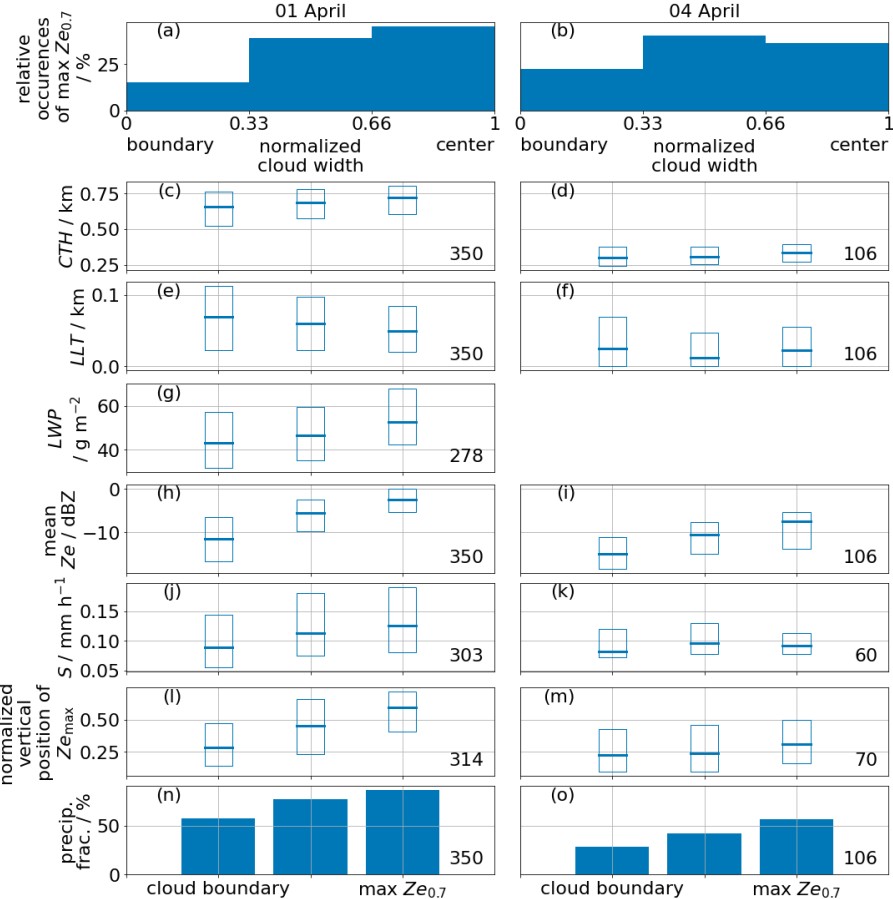

**Figure 7.** Relative occurrence of updrafts (max $Ze_{0.7}$) within all cloud streets that consist of at least 5 radar profiles on 01 April (a) and 04 April (b). Average composites of their microphysical cloud properties within the normalized distance between lateral cloud boundary and updraft on 01 April (left column) and 04 April (right column): cloud top height ($CTH$; c, d), supercooled liquid layer thickness ($LLT$; e, f), liquid water path ($LWP$; g), mean radar reflectivity ($Ze$) over each profile (h, i), snowfall rate at 150 m ($S$; j, k), vertical position normalized by the hydrometeor depth of the maximum $Ze$ for each profile (l, m), and fraction of precipitating profiles (precip. frac.; n, o). The median (horizontal line) and lower and upper quartile (box edges) are displayed at the boundary of the clouds, the updraft position, and in between.

## 4.2 Impact of roll circulation on cloud properties

To characterize how up- and downdraft regions of the roll circulations affect cloud properties, we build composites of the identified roll clouds (Sec. 3.2). For this purpose, we group cloud properties according to their distance from the maximum updraft region ($Ze_{0.7}$). First, we investigate the location of $Ze_{0.7}$ within a cloud of the 'cloud street' regime (Fig. 7a). Generally, the roll circulations on 01 April evoke clouds that typically form centered around the updraft of the circulation: most frequently (46 %) maxima of $Ze_{0.7}$ occur at cloud center and only rarely close to the lateral cloud boundary (Fig. 7a).



Cloud properties of cloud streets are composited for their relative distance to $Ze_{0.7}$ in Fig. 7c–o. On 01 April, the median of several parameters has trends though their variability, as expressed by the interquartile distance, is high: $CTH$ increases by 14 % (Fig. 7c) from cloud boundary until the location of maximum $Ze_{0.7}$. Consistent with cloud formation within updrafts, $LWP$ (23 %; Fig. 7g) and the mean of $Ze$ over each profile (81 %; Fig. 7h) increase. In contrast, a decrease of $LLT$ (29 %;
7e) by 20 m can be seen, which exceeds the error in $CTH$ of 10 m. This could be explained by the transport of ice particles by the updraft into higher parts of the clouds, thus increasing the mixed-phase region at the expense of the liquid layer.

The distribution of $S$ widens inside the clouds because particularly large $S$ evolve more extreme, while the distributions of $LLT$ and mean $Ze$ become more narrow. Precipitation events do not only intensify at the updraft location, but with 87 compared to 58 % of the profiles, also more profiles contain precipitation than at cloud boundaries (Fig. 7n). On 01 April, we
expect that most ice, indicated by $Ze_{\max}$, occurs at 0.6 of $D$ for updraft positions (Fig. 7l). At the cloud boundaries, most ice crystals are located at the lowest third of $D$.

On 04 April, updrafts form least likely at the cloud boundary as well (23 %; Fig. 7b), however, less symmetrical around the center as on 01 April. Moreover, the $CTH$ increases in the updrafts by only 7% (Fig. 7d). The increase rate of $S$ (Fig. 7k) and mean $Ze$ (Fig. 7i) within the clouds is about a factor of two lower. Moreover, $LLT$ stays constant within the clouds (Fig. 7f).
$LWP$ is not investigated here because the low values on 04 April are close to the detection limit of the measurements. At cloud boundary, the height with most ice within the cloud is similar to 01 April, but it increases by only 37 % at updraft position (Fig. 7m). The precipitation fraction is lower, i.e., only half at cloud boundary, but the increase by 34 percentage points is similar to 01 April (Fig. 7o).

The composites of the two cases show specific cloud property characteristics that can be used to evaluate high-resolution
model simulation runs. Shupe et al. (2008) found conditions similar to the stronger case on 01 April with an increase of $CTH$ and $LWP$ at updrafts of Arctic MPCs during non-MCAO conditions over land. For this case, $LWP$ reached much higher values of about $140\,\mathrm{g\,m^{-2}}$ for low $LWP$ cases, and their layer thickness of pure and mixed liquid was about 300–500 m with larger values at updraft locations. For their time series, the thickness of the liquid-only layer decreased in the updraft similar to our findings. For MPCs over sea ice observed during SHEBA, the ice water content peaked in altitudes of 0.6 of the cloud
depth (Shupe et al., 2006).

### 4.3 Development along fetch

To investigate how the open water surface affects roll circulation and cloud properties, we analyze their frequency of occurrence and evolution over all observed fetches (Fig. 8). We focus on roll circulation morphology first and use wavelength ($\lambda$) and aspect ratio ($AR$) as metrics (Sec. 3.2). Here, we only concentrate on the 'cloud street' regime where the roll circulation is well
developed.

On 01 April, the vertical extend of the roll circulation as indicated by $CTH$ is with a median of 700 m more than twice that high than on 04 April (300 m; Fig. 8a, b, I). On both days, $CTH$ clearly increases with fetch. The growth rate is by a factor of more than two higher on 01 April, however, a stagnation at 100 km fetch can be observed on that day. On both days, the frequency of occurrence and median of $\lambda$ (around 1.2 km) are similar (Fig. 8c, d, II). Generally, $\lambda$ increases slightly with fetch





on 01 April and reaches $2\,\mathrm{km}$ after $150\,\mathrm{km}$ approaching the width of the cloud streets seen by MODIS ($2\,\mathrm{km}$; Fig. 1). On 04 April, $\lambda$ stays roughly constant and aligns with the cloud street width of the MODIS images ($1\,\mathrm{km}$; Sect. 4.1). Regarding mean values, $\lambda$ is similar on both days, while $CTH$ is roughly a factor of two larger on 01 April. Consequently, the median $AR$ (Fig. 8III) is smaller on 01 April at 1.8 than on 04 April at 3.9. On both days, $AR$ stays constant with fetch at large fetches, while it decreases at small fetches (Fig. 8e, f). This decrease is strengthened on 04 April by the unexpectedly high $AR$ values that exceed 10. These values are provoked by weak convection at small fetches that are no roll circulations yet.

Next, we investigate the cloud properties as a function of fetch. Cloud cover starts to strongly increase at small fetches as can be seen in radar (black) and lidar (blue) measurements (Fig. 8g, h). Here, horizontal cloud cover is defined as the ratio between the number of cloud-containing and all profiles temporally averaged over one minute. The increase is strongest at $10\text{–}20\,\mathrm{km}$ fetch, and nearly complete cloud cover appears for larger fetches. The cloud cover from the radar is on average $4.5\,\%$ higher than that of the lidar, which might come from the radar's coarse horizontal resolution. The few cloud-free areas are small since $85\,\%$ of the observed cloud-free areas are less than $0.5\,\mathrm{km}$ wide (Fig. A2a).

Further, we investigate how exposure to open water influences cloud microphysical parameters. On 04 April, $90\,\%$ of the profiles containing liquid-topped cloud streets have $LLT$ smaller than $100\,\mathrm{m}$, which is more than on 01 April ($70\,\%$) when thicknesses of up to $500\,\mathrm{m}$ are observed (Fig. 8i, j, IV). Consequently, median $LLT$ is higher on 01 April ($75\,\mathrm{m}$) than on 04 April ($50\,\mathrm{m}$). No clear trend with fetch is evident.

The $LWP$ distributions of cloud streets differ between both days (Fig. 8V). On 01 April, the retrieved $LWP$ quartiles range from around 40 to $60\,\mathrm{g\,m}^{-2}$, the median is slightly below $50\,\mathrm{g\,m}^{-2}$, and thus $LWP$ is generally higher than on 04 April (interquartile range $0\text{–}25\,\mathrm{g\,m}^{-2}$; median $12\,\mathrm{g\,m}^{-2}$). Note that the maximum uncertainty in $LWP$ is estimated to be $30\,\mathrm{g\,m}^{-2}$ (Sec. 2.1). On average, $LWP$ increases with fetch (Fig. 8k, l). On 01 April, $LWP$ is more constant for fetches between 70 and $120\,\mathrm{km}$ and even decreases for fetches larger than $140\,\mathrm{km}$. This decrease results in a $50\,\%$ smaller total increase of $LWP$ over all observed fetches compared to 04 April, even though the absolute values are generally lower on 04 April.

On 01 April, the median of the mean radar reflectivity $Ze$ over each cloudy cloud street profile is around $-7\,\mathrm{dBZ}$ (Fig. 8VI). With around $-12\,\mathrm{dBZ}$, this value is lower on 04 April. Mean $Ze$ increases with fetch mainly within the first $70\,\mathrm{km}$ on both days, whereby the distribution is more narrow on 04 April. Most interestingly is the more frequent occurrence of values above $-5\,\mathrm{dBZ}$ on 01 April as they indicate snowfall.

For precipitating cloud street profiles, the upper quartile of $S$ is slightly below 0.2 and around $0.1\,\mathrm{mm\,h}^{-1}$ on 01 and 04 April, respectively, however, the lower quartile ($0.75\,\mathrm{mm\,h}^{-1}$) and median ($0.1\,\mathrm{mm\,h}^{-1}$) are similar on both days (Fig. 8VII) The course of $S$ (Fig. 8o, p) follows the one of $Ze$, as $S$ is derived from it. With $0.07\,\mathrm{mm\,h}^{-1}\mathrm{km}^{-1}$, the increase rate is similar on both days. Differences between the days are a larger variability within each fetch bin on 01 April and a shift of $S$ by $10\,\mathrm{km}$ towards larger fetches on 04 April. Hence, precipitation starts forming at fetches of 26 and $39\,\mathrm{km}$ on 01 and 04 April, respectively.

For fetches larger than $160\,\mathrm{km}$ on 01 April, $CTH$, cloud cover, $LWP$, mean $Ze$, and precipitation fraction probably decrease due to a remaining lee effect caused by Svalbard. Even though ERA5 reanalysis with its coarse resolution shows a rising air mass for cloud streets at these fetches, we suggest that the air mass subsides and thus forms low, non-precipitating clouds with



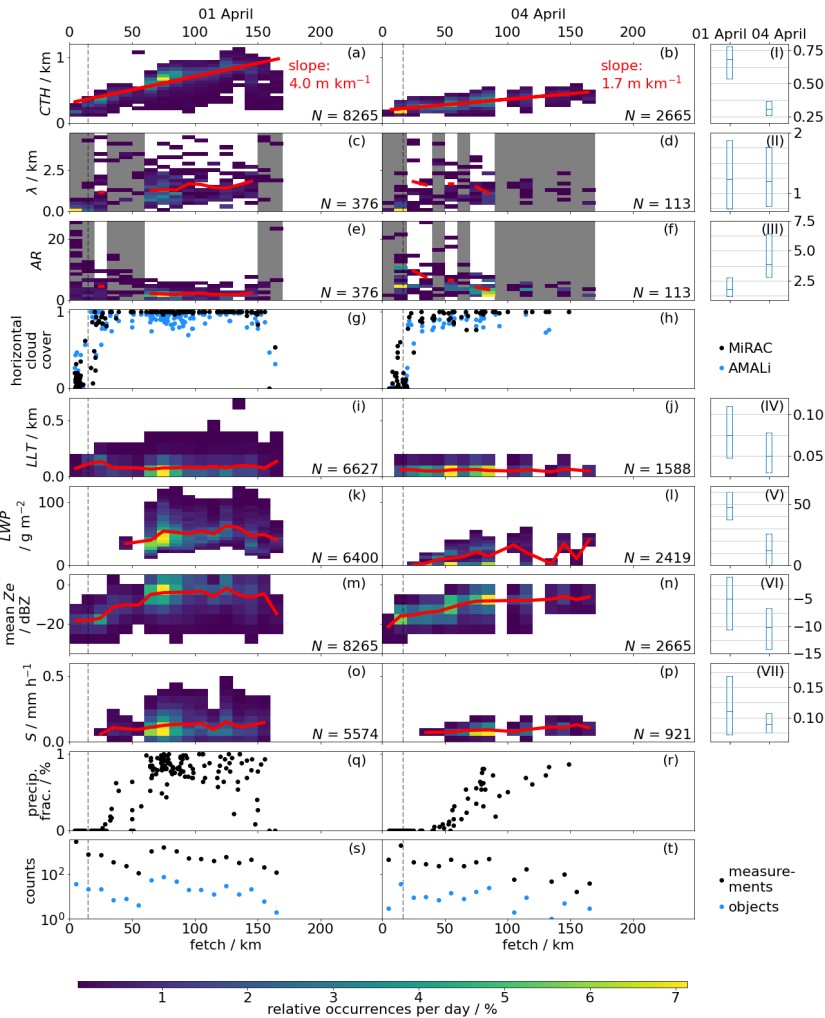

**Figure 8.** Development of geometrical and microphysical cloud properties with fetch on 01 April (first column) and 04 April (second column): cloud top height ($CTH$; a, b), wavelength of the circulation ($\lambda$; c, d), aspect ratio of the circulation ($AR$; e, f), horizontal cloud cover per minute (g, h) measured by the radar MiRAC (black) and lidar AMALi (blue), liquid layer thickness ($LLT$; i, j), liquid water path ($LWP$; k, l), mean radar reflectivity over each profile (mean $Ze$; m, n), snowfall rate at 150 m height for precipitating events only ($S$; o, p), precipitation fraction (q, r), and amount (s, t) of measured profiles (black) and identified cloud circulations (blue) per fetch bin. All $AR$ above 10 are caused by large $\lambda$ evoked by neighboring, cloud-forming convection that is not established as roll circulation yet. The red lines show the linear regression (a, b) and the course of the mean of the distribution for each fetch bin (c–f; i–p). The vertical gray lines (dashed) indicate the regime change from 'prior to cloud streets' to 'cloud streets' (Sect. 3.1). For robustness, bins of fetches with less than 10 roll circulation objects and of the 'prior to cloud street' regime are neglected to analyze the circulation (c–f; shaded). Boxplots (third column) show each distribution's median and interquartile range within the 'cloud street' regime (I-VII). The total amount of measurements ($N$) is given for each parameter and day.





small particles. In summary, all analyzed microphysical properties differ between the MCAO events, particularly $LWP$ and mean $Ze$. We observe slight changes in microphysical parameters with fetch. For most properties, the growth rates are similar on both days. However, the strength of the parameters is generally higher on 01 April when the $MCAO$ index is higher. The average $S$ with fetch on 04 April lags behind the one on 01 April by 10 km.

It has to be stressed, that the observed circulation and cloud evolution is not universal. As for our two cases, previous studies
by Brümmer (1999), Murray-Watson et al. (2023), Gryschka et al. (2008) or Tornow et al. (2021) have shown that clouds in MCAOs are highly variable and depend on the initial conditions. Nevertheless, some of our findings align with other case studies. For low cloud cover over sea ice, Brümmer (1996) found an increase in $CTH$ and cloud cover with fetch similar to our results. Our median $\lambda$ values align with $\lambda$ values obtained during the KonTur experiment (Brümmer et al., 1985; Markson, 1975; Brummer et al., 1982), ARKTIS '88 (Brümmer et al., 1992), '91 and '93 (Brümmer, 1999), and MIZEX (Walter and
Overland, 1984), i.e., 1.2 to 7 km at 250 km fetch. Similar to the 01 April, previous studies showed an increase of $\lambda$ with fetch (Brümmer et al., 1992), e.g., Brümmer (1999) found a wavelength of 1.3 and 3.1 km near sea ice edge and 140 km downwind, respectively. Except for some outliers, particularly on 04 April, the $AR$ of both days are similar to the observations during KonTur and MIZEX where a maximum of 4.6 was observed (Hein and Brown, 1988). Hein and Brown (1988) derived the $AR$ of the circulation from airborne gust probe measurement and also found that $AR$ decreases with fetch from 3.3 at
100 km downwind of the ice edge to 2.6 at 450 km. Other studies found both, decreaseing (Atkinson and Wu Zhang, 1996) and increasing $AR$ with fetch (Brümmer, 1999; Brümmer et al., 1992). However, these observations did not focus explicitly on small fetches as we do.

Averaged over liquid MCAO clouds between 2008 and 2014, the evolution of cloud cover with fetch obtained by MODIS (Murray-Watson et al., 2023) is in line with our observations. With 91 % at 35 km fetch, MODIS obtained only a 4 % smaller
cloud cover than averaged over both MCAOs of this study. However, their retrieved $LWP$ of 60 g m$^{-2}$ at sea ice edge and 130 g m$^{-2}$ at 430 km fetch is higher than during our cases. However, note that the satellite climatology explicitly excluded MPCs, which may explain the differences. The fact that MPCs are not studied explicitly by MODIS yet highlights the need for MPC observations.

## 5 Impact of riming

Ice growth has implications on precipitation and thus on the lifecycle of MPCs (Korolev et al., 2017). Therefore, we investigate whether roll circulation influences riming within the cloud streets and whether riming evolves with fetch. In doing so, we use a subset of in situ and remote sensing data during which *P5* and *P6* were collocated (Sect. 2). The data cover parts of the 'cloud street' regime only. To determine the degree of riming, we calculate the normalized rime mass ($M$) defined as the rime mass divided by the mass of the size-equivalent spherical graupel particle. Two methods following Maherndl et al. (2023b) are
applied. The combined method uses the closure of in situ particle size distributions and $Ze$ simulations obtained from running averages of in situ particle size distributions over 30 s. The in situ method relates $M$ to in situ particle shape measurements only. The results of both retrievals are comparable. However, since the collocation of *P5* and *P6* measurements might be inaccurate,



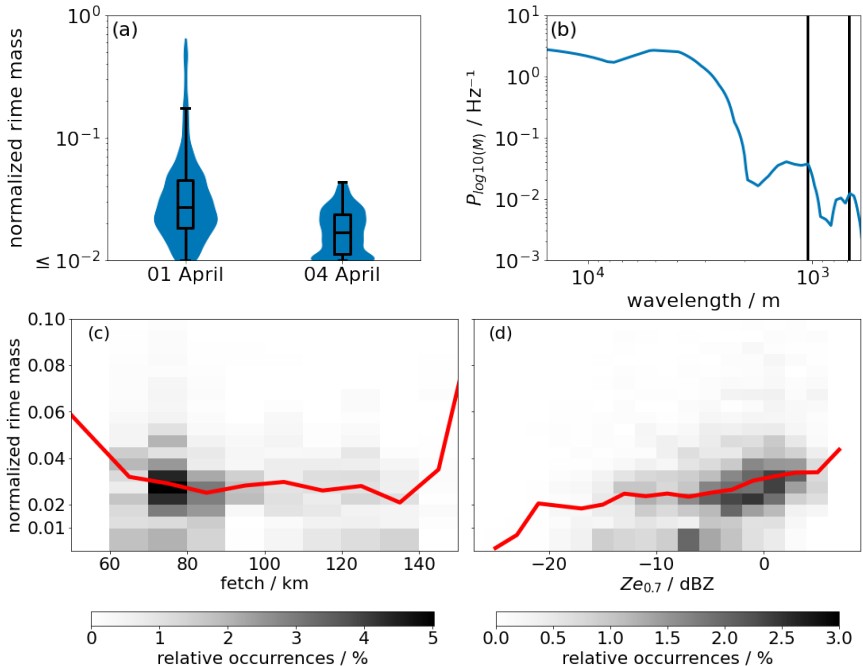

**Figure 9.** Normalized rime mass ($M$) obtained by the in situ method for collocated flight data of cloud streets on 01 and 04 April (a). The distribution of $M$ is represented by a violin and box plot, where the horizontal line marks the median. The respective power spectrum of $M$ on 01 April is shown in (b) with black lines marking important local maxima of the spectrum. Dependency of the normalized rime mass ($M$) obtained by the in situ method for cloud streets on 01 April on fetch (c) and updraft motion that is represented by $Ze_{0.7}$ (d). The red lines indicate the median of the $M$ distribution for each fetch or $Ze_{0.7}$ bin.

we only analyze $M$ of the in situ method for which no matching is necessary for the following. Note that our definition of updrafts might bias the following findings.

Considering particles with a rime mass fraction above $10^{-2}$ as rimed, more rimed particles exist on 01 April (97 %) than on 04 April (80 %). With a median $M$ of $10^{-1.6}$ on 01 April and $10^{-1.8}$ on 04 April (Fig. 9a), riming is only significantly active in cloud streets on 01 April. In the following, the variability of $M$ is investigated for 04 April. The spatial variability of riming is investigated by linearly detrended and mean-centered power spectra of $M$ obtained during the seven collocated segments on 01 April (Sec. 2). Edge effects are minimized by applying a Hann window for smoothing. Note that $M$ is not investigated

at updraft locations obtained from remote sensing observations from P6 because the vertical motions differ for the in situ observations from P6. Due to the units of variance, the power spectrum increases automatically for smaller wavelengths. The average of the power spectrum of $M$ peaks at about 0.7 and 1.1 km (Fig. 9b), thus, riming is variable at a spatial scale similar to the roll circulation (Fig. 8II). However, no general evolution of $M$ is observed with fetch (Fig. 9c). Instead, $M$ increases with $Ze_{0.7}$ (Fig. 9d). We suggest that riming is enhanced in updraft regions on 01 April.



## 6  Synthesis

The milestone of our study is the possibility of combining fine-resolved macro- and microphysical cloud observations in the initial MCAO transformation phase. In the following, we discuss the interaction between circulation, cloud macro-, and microphysical properties. Based on the two analyzed cases, the impact of the $MCAO$ index on these interactions is evaluated. The two scenarios and their different MCAO developments are illustrated by the schematic in Fig. 10. This schematic summarizes the remote sensing and in situ measurements presented in Fig. 4, 5, 7, 8, 9. Over sea ice, roll convection already forms, however, cloud streets have not developed yet. Prior to cloud street formation, the horizontal cloud extent and, with at most 10 %, the horizontal cloud cover are small. The clouds have no separate liquid layer at cloud top. Mean $Ze$ is relatively low, thus, the cloud particles are small and do not precipitate.

With fetch, the conditions develop: the averaged air temperature increases due to the presence of warm ocean water. This reduces the cooling rate of rising air masses and enhances the vertical extent of the largest turbulent eddies being ABL. For close locations, $CTH$ is slightly smaller than $BLH$, which indicates that the ABL caps clouds. Hence, $CTH$ increases with fetch. Roll circulations result in cloud streets after about 15 km fetch that are mostly MPCs.

The general evolution of cloud street properties within roll circulations is as follows: at the location of the strongest updraft, ice and liquid particles are lifted highest, which increases $CTH$, makes depositional growth of ice particles due to the colder temperatures more effective, and, thus, lifts the height with the highest ice mass within the hydrometeor depth. Due to larger ice particles over the whole cloud column, mean $Ze$ and lowest maximum $Ze$ increase, and the fraction of precipitation increases by roughly 30 percentage points. Due to the increased particle sizes, $S$ averaged over all precipitation events, and particularly high $S$ events increase. Descending or weak-rising motions next to the strongest updraft decrease the ice particle size and hence the averaged $Ze$. Note that high mean $Ze$ might occur at boundaries of clouds with overall similar-strong updraft motions, where air descends mostly next to the cloud.

Within the 'cloud street' regime roll circulations and clouds are non-static but rather develop with fetch: at small fetches, some convection is not established as roll circulation yet, which increases $\lambda$. Due to large $\lambda$ and relatively small $CTH$ at small fetches, $AR$ decreases logarithmically with fetch. For larger fetches and strong roll circulations, $AR$ is constant with fetch. This confirms the model by Brown (1972) stating that $AR$ increases for little energy available from convection. The width of cloud streets increases with fetch leading to the strongest increase of cloud cover at 10–20 km fetch, i.e., the initial cloud street formation, after which it is just below 100 %.

While the evolution of thermodynamics, clouds, and roll circulations with fetch is similar on both days, different thermodynamic conditions modify the intensity of the parameters. Over sea ice, conditions are only slightly different: on 01 April, $Ze$ values are a little lower likely due to drier conditions inside the ABL. $CTH$ values are lower on 04 April when dry air is advected above $BLH$.

Over water on 01 April, cloud top temperatures are colder than or at the low end of the temperatures within the dendritic-growth zone (DGZ; $-20$ to $-10$°C), where aggregation is a very dominant process (Chellini et al., 2022). This explains the high normalized rime mass. Contrarily to increasing $MCAO$ indices from rising $SST$, near-surface air mass subsides at 75





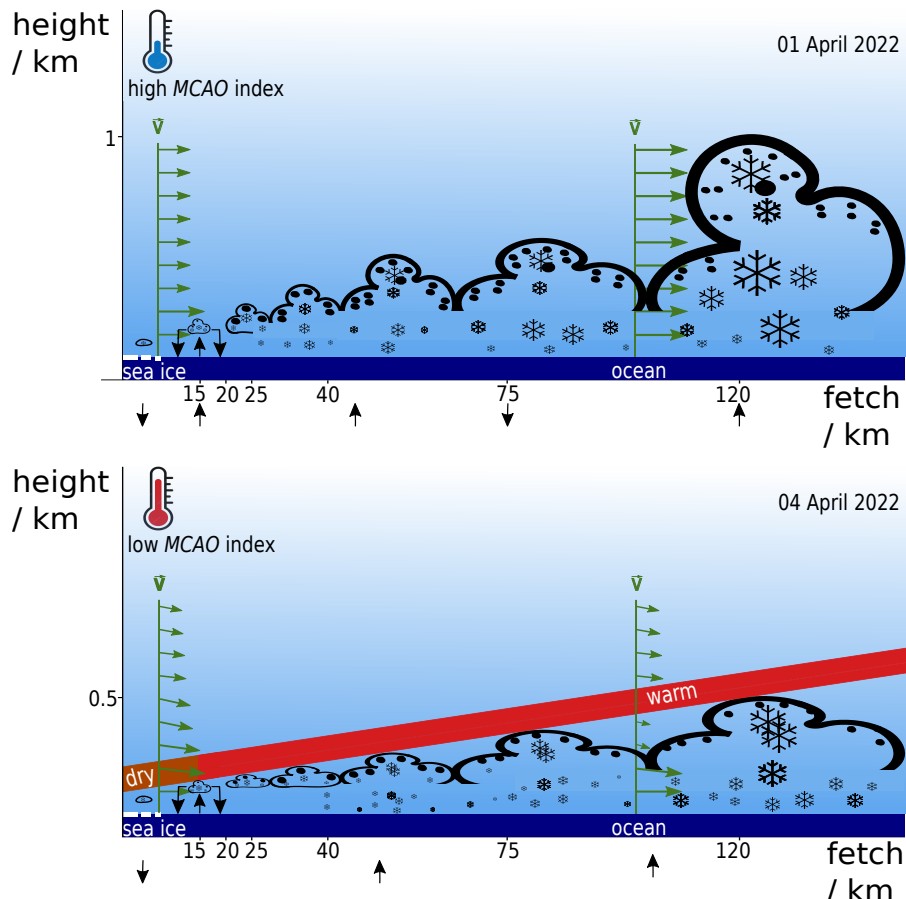

**Figure 10.** Sketch of the development of roll circulations and microphysics of the associated cloud streets with fetch on 01 April (first row) and 04 April (second row). The arrows at the bottom indicate the direction of the near-surface vertical motion.

until 120 km fetch on 01 April, which reduces the increase of $CTH$, mean $Ze$ and $LWP$ with fetch. Within the clouds,

510   $LLT$ decreases in updrafts because $CTH$ from lidar and radar are about 60 and 80 m higher compared to cloud boundary, respectively. Due to their size and shape, ice particles might be lifted more strongly than liquid droplets and entrain parts of the pure liquid layer converting it into a mixed layer. Since the liquid is exposed to more ice particles, more liquid droplets might be depleted during riming. We expect this enhancement of riming in updrafts because $M$ correlates with $Ze$. Riming increases ice particle size, $Ze$, and $S$ in updrafts. The $LWP$ increase in updrafts might indicate that condensation is more favored than

515   depletion of liquid.

   On 04 April, riming is not active because the cloud top temperatures lie within the DGZ where aggregation is favored. Within the clouds, less lifting indicated by lower $MCAO$ indices lowers the altitude of most ice with regard to hydrometeor depth particularly at the strongest updraft by half compared to 01 April. Additionally, $LLT$ stays constant because liquid and ice particles are lifted by the same amount. $Ze$ and $S$ increase in updrafts probably due to large dendrites that form by aggregation.




The reduced buoyancy and warm air advection above $BLH$ diminish the vertical extend of the ABL, and thus the growth rate
of $BLH$ and $CTH$ with fetch by roughly 55 % compared to 01 April. $AR$ is higher on 04 April because smaller $CTH$ are not
compensated by $\lambda$, which on average stays constant. Particularly at small fetches, weak convection increases $AR$ confirming
Brown (1972). On 04 April, more shallow clouds exist with fewer parts of possible supercooled liquid and warmer supercooled
liquid layer temperatures. Less supercooled liquid reduces the amount of liquid-topped cloud profiles, $LWP$ and $LLT$. The
temperature, low amount of supercooled liquid, and weak convection prevent riming and reduce $S$ and mean $Ze$ even with
favored aggregation. All these mechanisms might delay the precipitation onset on 04 April by more than 10 km.

## 7   Conclusions

Our study investigates the evolution of thermodynamics, cloud/circulation morphology, cloud microphysics, and riming for the
first 170 km fetch of two MCAO events. Previous studies resolved the initial phase of MCAO transformation, which determines
the subsequent evolution, only coarsely and did not combine micro- with macrophysical cloud and circulation properties. We
investigate high-resolved airborne remote sensing observations of two MCAOs observed during the HALO-(AC)[3] campaign
in a quasi-Lagrangian way using back trajectories. We establish a novel approach to detect roll circulations from vertical radar
profiles only, allowing us to answer the posed research questions (Sec. 1).

I. Which thermodynamic conditions characterize the two MCAO events?

The two events on 01 and 04 April feature northerly winds advecting dry and cold air masses into the Fram Strait. The
event on 01 April shows colder air temperatures leading to a factor of two stronger $MCAO$ index and stronger surface
fluxes. Wind shear induced by wind speed (01 April) and wind direction (04 April) is present.

II. Which cloud properties are associated with roll circulation?

A composite analysis reveals that, compared to cloud boundary, updrafts increase cloud top heights, liquid water path,
snowfall rate, and the height within the cloud containing most ice. Regarding median properties, cloud top height, liquid
layer thickness, liquid water path, and snowfall rate are roughly a factor of two lower for the weaker MCAO on 04 April.

III. How do circulation and cloud properties change with fetch in the initial state of MCAOs and when do cloud streets start
to precipitate?

An increase with fetch is found for boundary layer and cloud top height, temperature and humidity of the boundary
layer, liquid water path, radar reflectivity, near-surface precipitation rate, horizontal cloud cover, and the fraction of
precipitating profiles. The aspect ratio of the circulation decreases with fetch for small fetches and stays constant for
larger fetches. The studied cloud streets start precipitating at 25–30 km.

IV. What is the impact of riming on MCAO transformation?

For riming to be active, it has to be colder than -20°C. On 01 April, maxima in the normalized rime mass have a horizontal
spatial scale similar to the roll circulation but do not depend on fetch.

We established composite approaches to characterize the roll circulation and fetch dependence (Fig. 7 and 8). Such metrics
can also be generated from high-resolution model output and used to evaluate their performance. In particular, it will be





interesting to analyze whether such models successfully reproduce the difference between both cases and whether the observed factor of two scaling appears.

It would have been interesting to characterize the impact of the marginal sea ice zone (MIZ) on the air mass transformation. However, too few data with variable fetches due to different fetches over MIZ only exist. To study the impact of the sharpness of MIZ and flow divergence on cloud evolution, more observations at constant fetches over open water and variable fetches over MIZ near the sea ice edge must be obtained in the future. Moreover, since the observed cloud top temperatures lie within the dendritic-growth zone, aggregation would be another interesting process to study, which is possible by dual frequency radar

observations (Chellini et al., 2022). The Clouds over cOMPlEX environment (COMPLEX) campaign will raise the opportunity to study clouds near sea ice edge where an airborne W- and G-band radar will be operated northwest of Svalbard in spring 2025.

*Data availability.* Processed radar, in situ, and dropsonde observations, as well as retrieved $LWP$ and $CTH$ data from the HALO-(AC)3 campaign are currently being prepared for publication on PANGAEA. The regarding DOIs will be updated during the review. All airborne

data are accessed via the ac3airborne module (Mech et al., 2022b). The merged MODIS-AMSR2 sea ice concentration data are provided by the Institute of Environmental Physics at the University of Bremen (Ludwig and Spreen, 2023). Raw in situ data are stored at the German Aerospace Center and available on request. Back trajectories are calculated from ERA5 reanalysis data (Hersbach et al., 2017, 2020).

## Appendix A: Error of fetch induced by resolution limitations

The temporal resolution of the calculated fetch is 1 min. We calculate back trajectories from ERA5 wind fields with a horizontal

resolution of roughly 30 km. For an average flight speed of $80\,\mathrm{m\,s^{-1}}$, *P5* travels a distance of about 5 km during one minute. Taking the same trajectory for all measurements within one minute thus does not reduce the spatial resolution. For SIC, we take merged MODIS-AMSR2 data with a horizontal resolution of 1 km (Ludwig et al., 2020). The ERA5 wind field is relatively homogeneous. Thus, neighboring trajectories do not differ significantly (Fig. 1e, f). Differences in fetches between two neighboring trajectories mainly come from differences in SIC along the trajectories. The median of the relative change

between two adjacent fetches is 9.6 %.





**Table A1.** Sensitivity of the steps of the algorithm applied to identify roll circulation objects. Relative changes of the number of objects in total, number of objects inside the 'cloud street' regime, cloud top height ($CTH$) of cloud streets, and aspect ratio ($AR$) of the roll circulation to the results obtained by the applied configuration after adjusting, i.e., mostly doubling, parameters.

| modification | total number of objects | number of objects within 'cloud street' regime | $CTH$ | $AR$ |
|---|---|---|---|---|
| III: average over 6 s | -17.8 % | -20 % | +24 % | +23 % |
| IV: 0.6 of hydrometeor depth | -1.8 % | -1 % | 0 % | 0 % |
| IV: 0.8 of hydrometeor depth | +3.4 % | 0 % | 0 % | 0 % |
| VI: width of 5.8 samples | -31.3 % | -37 % | +59 % | +57 % |
| VI: If $Ze_{\mathrm{back}} \geq 0.67\,\mathrm{mm}^6\,\mathrm{m}^{-3}$: prominence of at least $0.2\,\mathrm{mm}^6\,\mathrm{m}^{-3}$ | -9.8 % | -11 % | +10 % | +11 % |
| VI: If $Ze_{\mathrm{back}} < 0.67\,\mathrm{mm}^6\,\mathrm{m}^{-3}$: prominence of at least $1\,\mathrm{mm}^6\,\mathrm{m}^{-3}$ | -9.3 % | -10 % | +10 % | +11 % |





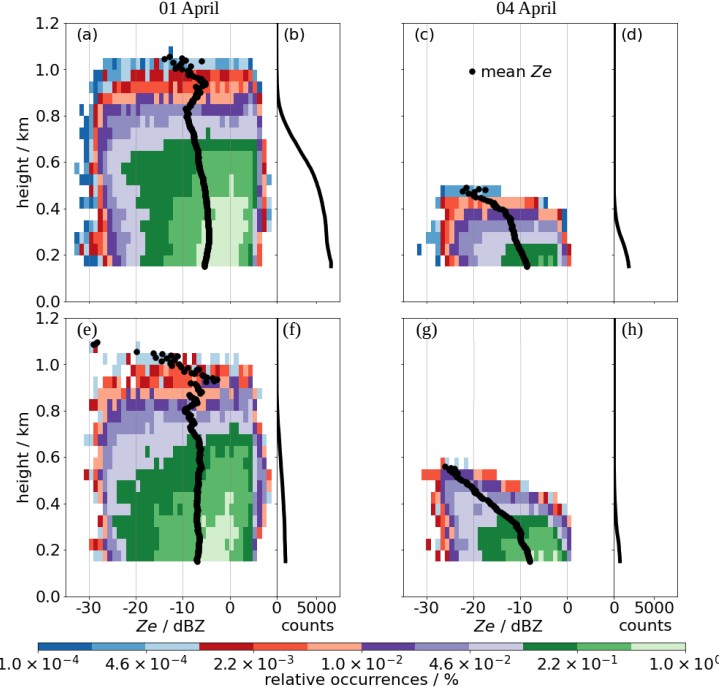

**Figure A1.** Contoured frequency by altitude diagram (left) and absolute counts per altitude (right) for liquid-topped (first row) and non-liquid-topped (second row) radar reflectivity ($Ze$) profiles obtained by MiRAC in the 'cloud street' regime on 01 April (a, b, e, f) and 04 April (c, d, g, h). Moreover, each mean $Ze$ profile (black dots) is displayed. The total amount of liquid-topped and non-liquid-topped profiles is 8979 and 1951, respectively.



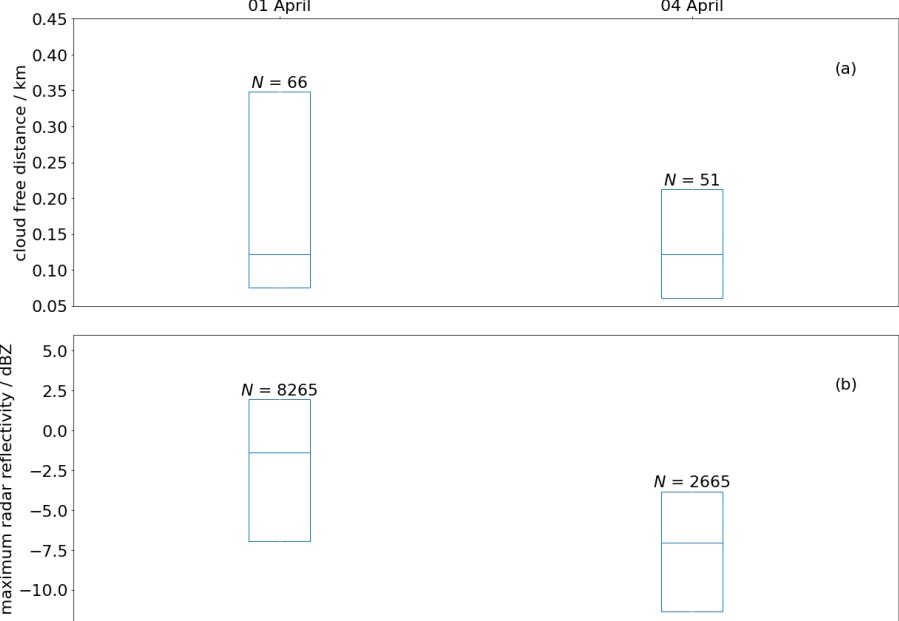

**Figure A2.** Boxplots for cloud free distances (a) and the maximum radar reflectivity of each profile for the 'cloud street' regime on 01 April (left) and 04 April (right). The boxes show the median and interquartile rannge, and $N$ the number of samples.

*Author contributions.* IS performed the analysis, visualization, writing, and developed the methodology. Together with SS, MK, AE, and SC, the paper was conceptualized and results were discussed. MM and AE developed the flight strategy for the cloud street investigation. BK calculated the back trajectories. NM collocated the *P5* and *P6* measurements and computed the rimed mass fraction. All authors contributed to manuscript revisions.

*Competing interests.* The authors declare no competing interests.

*Acknowledgements.* We gratefully acknowledge the funding by the Deutsche Forschungsgemeinschaft (DFG, German Research Foundation) - Projektnummer 268020496 - TRR 172, within the Transregional Collaborative Research Center "ArctiC Amplification: Climate Relevant Atmospheric and SurfaCe Processes, and Feedback Mechanisms (AC)[3]" - in subproject B03. Furthermore, we acknowledge the support for the Article Processing Charge from the DFG (German Research Foundation, 491454339). We are grateful for the support from the Alfred-
Wegener-Institute, DLR, and aircraft crews during the HALO-(AC)[3] campaign. Moreover, we acknowledge the use of imagery from the NASA Worldview application (Nasa Worldview, 2023a, b, c), part of the NASA Earth Observing System Data and Information System (EOSDIS). Furthermore, we thank the Institute of Environmental Physics, University of Bremen for providing the merged MODIS-AMSR2



sea ice concentration dataset (Ludwig and Spreen, 2023). Many thanks to the PIs Mario Mech, Stephan Borrmann, Johannes Schneider, and Veronika Pörtge. We are also grateful to Bjorn Stevens for discussing the flight strategy and cloud street investigation, Matt Shupe for discussing the roll circulation identification, and Vera Schemann for discussing future model evaluation efforts.





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
