# Peer review of "Clouds and precipitation in the initial phase of marine cold air outbreaks as observed by airborne remote sensing"

_EGUsphere, 2024_

## Referee Comment (RC2)

Review of Imke et al. 2024
Clouds and precipitation in the initial phase of marine cold air outbreaks as observed by airborne remote sensing doi: 10.5194/egusphere-2024-850

This manuscript focuses on 2 cold-air outbreaks observed during the AC3 campaign in April 2022 and describes their characteristics. The main motivation presented for the study is that the observations provide a valuable reference for future modeling studies.

This is a useful study, I do have a fair number of comments I'd like the authors to address.

Abstract: Why these 2 CAOs in particular? Are they representative? Where do they fall within the more comprehensive assessment of Murray-Watson for example? This is addressed later in the document, but would be good to bring into the abstract. Is there a new process insight or hypothesis developed from their study that is new to their understanding? It is not clear to me why modelers should choose these examples above others, other than the opportunistic sampling. The authors also don't come back to this argument towards the end of the document.

It makes sense to me that the 2nd CAO is the weaker of the 2, as it occurred 3 days later, and likely experienced similar synoptic conditions. The authors treat the 2 CAOs as if they are independent of each other. What initial conditions did they share? A figure indicating the dominant synoptic situation and how it evolved would helpfully complement the first figure. I hypothesize a coastal low interacted with the northerly CAO flow to develop that convergence line, for example.

In addition, the English is awkward in places. I recognize English is not the first language of any of the authors. I will point out specific places in the more detailed comments below.

More detailed comments:
Abstract, line 12: "the liquid layer is wider" is unclear here. Do you mean a larger horizontal dimension of the rolls? Except the previous sentence says the width of the roll circulation is similar between the 2 cases. "Occurrence" -> occurrence of what? "are" -> "have"

Line 30: "them" what is 'them'? Define

Lines 33-36: unclear to me how the 2nd sentence relates to the first. Isn't the first sentence saying vapor depositional growth is favored when the vapor pressure> ice and liquid saturation?

Line 37: awkwardly written.

Line 73: "were" -> "was"

Line 83: "extend" -> "extent". Same for line 391. And line 520.
Line 85: awkward. Overall the sentence is vague. What are the induced changes in dynamics and clouds?

Line 160-161: how do you know most of the liquid is near the top? As written this sentence sounds presumptuous. And what is the motivation from deriving LWP from the slanted profiles? Superficially the nadir profiles would make more sense.

Line 240: wouldn't the strongest updrafts generate liquid drops that are less obvious to the radar than the radiometer? Why expect ice production and liquid production to be collocated?

Line 256: 2.9 samples = what horizontal distance?

Fig. 4: this is the first introduction of the measurement regimes, so you may want to identify them in the caption, it's hard to understand this figure otherwise. Are these the same as provided in Table 1? the legend and symbol shapes are hard to see. Besides increasing them in size, would also suggest increasing the size of the overall figure.

Line 281: what is the uncertainty in the LLT metric?

Line 290: "precipitating" -> precipitation

Section 4.1: there really should be a synoptic analysis and figure provided, this would be a good spot for it.

Line 296: was VIIRS imagery, of higher spatial resolution, not available?

Table 2 is nice. Does the BLH top correspond to the cloud top, or the mixed-layer height? Indicating the range of cloud top heights would be useful also.

Lines 301-304: fig. 4 is difficult to follow to be honest. The dropsonde and ERA5 values don't agree well, the disagreement is more than I would have expected based on Seethala et al. 2021.

Line 308: I don't see mean thermodynamic profiles provided as part of fig. 1e,f,…..these figures only indicate the locations of the dropsondes.

Line 303: One also can't see from Fig. 4 that the MCAO follows the SST (SST isn't shown) and basically we are just seeing a few values for the ERA5 dataset, because of the coarse resolution. I would suggest increasing the size of the dropsonde-derived values in fig. 4, and including SST, and reducing the size of the ERA5 values. I am confused why there are so many ERA5 values. The interesting aspect of this figure is the sharp increase in the MCAO index for 1 April (isn't it)? How much of the fetch is over the MIZ? It would be useful to include the beginning point in the figure 4.

Line 314: where are you showing the surface flux values you refer to?

Line 315: here you define the BLH. This would be useful to include in the caption of table 2 as well.

Lines 328-330: I rather think a spatial plot of the ERA5 winds and Geopotential height would be more informative here.

Line 339: that the clouds only reach 1km and 500m respectively is interesting too. The CAOs documented at COMBLE tend to reach higher.

Line 342: awkward writing. Line 343: you could look at the near-surface RH to examine if more evaporation is happening.

Line 347: not convinced this is the best place to mention Morrison et al 2012. That is a modeling study based on more weakly-forced clouds. If you are going to mention it, better left to a discussion later on.

Fig. 7: this is a nice figure. The caption should state clearly what the 3 groupings represent. Is is the same for all panels? The lowest one is labeled differently. How wide/coarsely resolved are these distances? I just wonder if ice production is truly colocated w liquid production, or perhaps they get lumped together through a coarseness in the grouping.

Line 367: not following this sentence. What do you mean by extreme?
Line 368: 87 -> 87%
Line 370: spell out what '0.6 of D' means (it's easy to lose track of the acronym meanings).
Line 375: is the measurement limit known to the reader somewhere earlier? I don't recall seeing it.

Line 380: I think the clouds looked at within Shupe 2008 are different enough from CAOs that this analogy isn't quite right. It could just be a coincidence. You could say in the discussion section that examining if CAO clouds are a truly unique cloud regime that behave differently from non-CAO MPC is a topic worthy of future study (I suspect the differences in surface fluxes do genuinely separate CAO and non-CAO clouds).
Lines 381-383: if you want to keep this, it would be better suited for a discussion section towards the end.

Line 398-399: vague. Might help to indicate the distances of 'large' and 'small' fetches.

Line 407: to me it makes more sense to look at how cloud microphysics depends on the surface fluxes, as opposed to the exposure to open water. You can actually estimate the surface fluxes from the dropsondes, this was done within Seethala et al 2021. It would be interesting to know those values.

Line 408-409: this sentence doesn't make sense.

Line 413: here is where I think the reader learns the LWP uncertainty. Better to say the LWP values are beneath the maximum uncertainty of 30 g/m2.

Lines 427: how do you deduce the subsidence? From satellite imagery?

Lines 434- : I appreciate the effort to place the 2 CAOs into context and not claim the noted features are universal (though they well might be). It would be nice to see this sentiment expressed in the abstract as well.

Line 445: decreaseing -> decreasing

Line 452: Khanal et al 2020 examine MPCs using MODIS, cited below.

Line 475: since updraft speeds are correlated to surface fluxes, you could examine the dependence of riming on dropsonde-derived surface fluxes to put this statement on a stronger footing. Unless you have in-situ vertical velocities from one of the planes (it doesn't seem like it though).

Line 485: the -> these, before 'conditions'

Lien 486: do you show CTH<BLH somewhere? I don't recall seeing that.

Line 508: isn't dendritic growth through vapor deposition, followed by aggregation, a different growth process than riming? I'm not sure I understand how aggregation encourages more riming.
Line 508: and how do you know SST is rising? You haven't shown SST anywhere.

Line 510: this is difficult to follow, not sure what this sentence is saying.
Line 511: I think liquid droplets are typically more readily raised, because they are smaller than ice particles. Or maybe you just need to be more precise about the particle size - but I'm not sure you have the microphysical measurements to back up this statement. All in all, this paragraph is too speculative. Please dial it back.
Line 516: you are contradicting what you said on line 508.

Conclusions: I like that you brought it back to the motivating questions.

Line 549: I don't believe temperatures have to be colder than -20C for riming to be active, this is certainly not universally true, nor do you explicitly show this.

References:

Seethala, C., P. Zuidema, J. Edson, M. Brunke, G. Chen, X.-Y Li, D. Painemal, C. Robinson, T. Shingler, M. Shook, A. Sorooshian, L. Thornhill, F. Tornow, H. Wang, X. Zeng, L. Ziemba, 2021: On assessing ERA5 and MERRA2 representations of cold-air outbreaks across the Gulf Stream. *Geophys. Res. Lett.*, **48**, doi:10.1029/2021GL094364

Sujan Khanal, Zhien Wang, Jeffrey R. French, Improving middle and high latitude cloud liquid water path measurements from MODIS, Atmospheric Research, Volume 243, 2020, 105033, ISSN 0169-8095, https://doi.org/10.1016/j.atmosres.2020.105033.

---

## Author Response (AR1)

**Reviewer 1**

Thank you very much for spending the time and effort to thoroughly review our study, which helped to improve the manuscript. Based on the reviewer's comments, we rewrote most of the manuscript and shortened it significantly. We refined the research questions and give more insights into how our measurements are suited for model evaluation. Specifically, we worked on the description of the synoptic conditions during the MCAO event and of how the two flights probed different flavors of it. In the following, we reply (blue) to all reviewer's comments (black). Text passages from the manuscript are in italics. In our answers, we always refer to the line numbers of the revised manuscript.

One change that does not correspond to a reviewer's comment needs to be highlighted first. We found a problem in the roll circulation detection algorithm. For radar reflectivity (Ze) profiles with more than one cloud layer, the hydrometeor depth of the upper cloud layer was used by the algorithm previously. However, a few radar profiles observe two cloud layers with a very short cloud-free distance in between the layers, which likely occurs due to limited radar sensitivity. For these cases, the methodology to derive a roll circulation "object" ignored the lower cloud part. Thus, the updated version of the algorithm counts two cloud layers as one as long as the cloud-free distance between them is thinner than 50 m (see line 147). This update increases the hydrometeor depth for these cases. However, this modification of the algorithm only leads to small changes. Previously, we identified 364 and 109 cloud circulation objects in the 'cloud street' regime on 01 and 04 April, respectively. This changes to 356 and 112 objects which is corrected in line 251 .

**Major comments**

First of all, the manuscript is much too long and contains too much detail not necessary for the reader to know. Much of this could just go all together, while some could be put in supplementary information.

We reduced the length of the manuscript by removing the synthesis section and moving the most important information into other sections for a better narrative. Moreover, we moved details about the methodology into the Appendix and reduced the amount of details in all sections, particularly in Sect. 4.4. Some extra figures can be found in a supplement.

Second, the discussion is sometimes very confusing, jumping from conditions in one of the cases to the other and back, and it is very easy to loose track on which case is actually discussed. It would be much better if at least initially, with all the background conditions, the two cases are discussed entirely separately.

We improved the flow of the manuscript by shortening it. Now, both cases are first discussed separately in Sect. 4.1 before focusing on similarities and differences between the two cases in the following sections.

Moreover, there is too much details in the figures, that are also much too small, and the figure captions are very disorganized. Figure captions should be very succinct and organized and – in contrast to the text – not have a narrative; just the bare bone facts about what is in the figure.

We reduced the details within Fig. 6 and 8 and better organized Fig. 2 and 3. We also divided Fig. 7 into two figures. We hope that this reduces the details in the figures and makes them easier to understand. Furthermore, we revised the figure captions so that they only describe necessary information.

Third, the author seems to work hard to make the case that the cases are really different, while what strikes me is how similar they are. Sure, magnitudes are very different but looking at – for example – trends along the fetch, I find the underlying structure surprisingly similar.

Similar and not similar highly depends on what type of clouds are compared. In our study, we focus on clouds in marine cold air outbreaks, which are of a specific nature, and general characteristics of clouds in MCAOs (trends along fetch) are well known. Thus, we state in line 408: „*The evolution of cloud microphysics with fetch is similar on both days, however, thermodynamic conditions modify the intensity of the parameters*.“ Note that we integrated more synoptic aspects into the paper following reviewer 2. This shows that both cases belong to one long-lasting MCAO event but have slightly different flavors. More details are provided in Sect. 2.1. In the conclusions, we also discuss that looking at two events with many commonalities but significant differences in cloud properties provides an interesting benchmark for testing models.

Finally, the conclusions and the answers to the research questions, given initially, are very trivial and reveal nothing new, that couldn't have been guessed given just the basic information.

We agree, that some of the conclusions might have looked trivial as these agree with the general concept of the cloud evolution in cold air outbreaks. However, our study is able to confirm theory by detailed airborne observations that were not available before in terms of instrumentation but also in terms of resolution close to the sea ice edge, i.e., the initial stage of cloud formation in MCAOs. We sharpened the reformulated research questions:

*I. What are the differences between the environmental conditions on both flight days and what are their implications on the cloud development?*

*II. Can we identify characteristic changes in cloud and precipitation properties perpendicular to cloud street orientation, i.e., within the roll circulation?*

*III. How do roll circulation, cloud, and precipitation properties evolve with fetch in the initial MCAO phase, e.g., up to travel times of four hours?*

Furthermore, we reformulated the introduction to emphasize that we provide unique data and evaluation metrics to test MCAO simulations which is further discussed in the conclusions.

**Minor comments:**

Line 35-36: Unclear; I don't understand why ice cannot grow at the expense of the liquid just because the humidity is above saturation w.r.t. ice.

We removed this statement.

Line 45: "flow divergence" how and where?
Line 51: Why – and where - does the fluxes increase with a sharper MIZ?

Both statements were removed when we shortened the introduction.

Line 64-65: While the open water distance passed by an air parcel increases with open leans and open water over the MIZ, it is unclear what this means in terms of fluxes; this depends on the water temperature which to some extent will be controlled by the neaby ice.

The text has been shortened here. We now also show SST and sensible heat flux from ERA5 and dropsondes in Fig. 3. Note, that in the marginal sea ice zone the sea surface temperature (fixed to roughly -2°C) is always higher than the ice temperature and thus open water always enhances turbulent heat fluxes compared to closed ice (e.g., Liu et al., 2006).

Line 66: What cloud-street charateristics are variable and how?

Removed when shortening the text.

Line 74: Unprecise; the spatial resolution of CloudSat along the track is quite good; however, the swats are far apart and nearby swats occur sparsely and swats are generally not aligned with CAOs.

The statement was removed when we shortened the text. Note, that coarse is always relative: in respect to our study the roughly 10 times finer resolution of the airborne measurements is able to resolve details within the cloud streets which is not possible by MODIS or Cloudsat.

Line 128-129: Unclear; I have a hard time seeing the convergence line discussed here.

We added a synoptic map to Fig. 1 and explicitly mark the convergence zone, which also corresponds to the cloud band of enhanced MODIS LWP.

Line 130: Maybe "different" instead of "varying"; the latter would be changes within each case while the former changes between the cases.

We use different.

Figure 2: An interesting feature in this figure is that the reflectivity seems not to be symmetric; the maximum seems to be shifted to the right (in the figure) and is not centered in between the downdrafts. This is not even discussed, and I have no idea if this was just spurious or is a systematic feature.

Yes, we see it as an advantage of our algorithm that no symmetry needs to be assumed as updraft and downdraft regions are diagnosed independently. We now discuss this fact in line 248 (*"According to our definition, the maximum updraft (maximum $Ze_{0.7}$) does not necessarily need to be centered between the two detected edges of our roll circulation object."*) and in line 345 (*"As explained before, objects are not necessarily symmetric. However, most clouds form centered around the updraft of the circulation: around 50% of the time, maxima of $Ze_{0.7}$ occur within the central tercile of the cloud and only rarely within the tercile closest to the lateral cloud boundary (7%)."*).

Line 160: Unclear what you mean by "below 30 g m-2". Do you mean the accuracy is "better than 30 g m-2" or that there is something special with the accuracy "at values below 30 g m-2"? Moreover, 0.5 30 g m-2 is also "below 30 g m-2", so if the first is true, it doesn't say anything.

The maximum uncertainty of LWP induced by the retrival process is below 30 g m$^{-2}$. See Ruiz-Donoso et al. (2020) for detailed information about how this uncertainty is understood. A detailed paper about LWP observations during the HALO-(AC)[3] campaign is in preperation. In the manuscript, we eliminated the part on the detection limit and reformulated the statement about the maximum uncertainty: "*Depending on atmospheric conditions, the maximum uncertainty is below 30 g m$^{-2}$ (Ruiz-Donoso et al., 2020).*" (line 156).

Line 164: I thing you mean "… non-constant manner." or maybe "… non-constant way."

Corrected.

Line 187: What does it mean that you use 850 hPa (rather than some lower value)? Both cases are rather shallow at the distances you study, so why not use e.g. 925 hPa.

Our definition of the MCAO index follows the definition of previous studies like Walbröl, et al. (2024), Dahlke et al. (2022), Papritz et al. (2015), and Kolstad (2017). We take the potential temperature at 850 hPa to be able to compare our MCAO indices with indices from other events analyzed in previous studies. Moreover, we want to be sure that the layer of advected warm air above ABL over open water on 04 April, which does not represent the temperature of the free troposphere, does not influence the MCAO index.

Section 3.2: For something as basic as a trajectory calculation, this section is exceeeeedingly long. It can be shortened to a third of its current length.

We shortened this section.

Line 204-209: This paragraph is very confusing and it is unclear what the argument made really is. Moreover, the assumption that the cumulative flux is independent of wind speed, since length and time are interchangeable, is based on an assumption that the temperature difference between air and sea surface is constant, which is almost certainly not the case.

We deleted this paragraph and just make the point that fetch and travel time can be analyzed interchangably for our data. For Fig. 8, we show both fetch and travel time to enable the reader to better compare with other studies. See line 206: *„For flows unaffected by land masses, travel time over open water and fetch can be linearly converted and are both valid to study.*"

Line 213: I don't know what and integrated time means.

We replaced integrated time by travel time over open water.

Line 215-217: I beg to differ; the IFS has significant biases in almost all boundary-layer parameters and especially in the turbulent fluxes. This is borne out by the quoted 200 m error in BL growth (line 217) which to me in these conditions is not a small error.

We removed the statement about the BLH as it does not fit the story line. Note, that most HALO dropsondes have been assimilated in ERA5 leading to an improved performance for our study case. This is now mentioned in Sec. 2.3, line 195. Thus, ERA5 sensible heat flux compare well with estimates from the dropsondes (Fig. 3).

Line 218: I don't understand the argument of this resampling, which provides trajectories at an almost ridiculous resolution, far exceeding the resolution in the input data from ERA5.

Our aim is to attribute a fetch to each airborne measurement having a resolution of 1 second. To do so, we do not resample the minutely resolved fetches but rather assign a fetch to all observations within the respective minute. We have rewritten the statement in the manuscript: *„Specifically, we calculate back trajectories for the previous 12 hours for every flight minute and assign them to the observations within each minute.*" (line 211). The aspect of resolution (and its limitation) is discussed in line 219: "*Note that due to the resolution of ERA neighboring trajectories are rather similar (Fig. 1g, h). Differences in fetches between two neighboring trajectories mainly come from differences in SIC along the trajectories. The median of the relative change between two adjacent fetches is 9.6%.*".

Line 231: How do you handle the SST in the MIZ?

In the paragraph the reviewer refers to, we only calculate the distance traveled over open water where the geometric effect of sea ice is taken into account not the SST. In Fig. 3, we also include a map of SST (from ERA5) to illustrate the SST in the region. In the MIZ, it is basically fixed to freezing point of sea water. Quantitatively, we only need SST for the calculation of the MCAO index from the dropsondes. For this purpose, AVHRR observations of SST are used (see Sect. 2.3). As we only calculate MCAO indices for the cloud street regime the MIZ is not affected.

Section 3.2: This is one of the most interesting introductory sections, but it is also much too long and detailed. Describe the principles and leave out the details, or move them to an appendix or supplementary information.

We reduced the details within this section and moved the details about the algorithm and the selection of the height threshold (0.7 of the hydrometeor depth) into the Appendix. To provide the reader with an overview we illustrate the principle of the algorithm by a simple flow diagram added to Fig. 2.

Line 244: Why 0.7? Are the results sensitive to this choice? Why not a constant distance into the cloud? Entrainment does not necessarily penetrate deeper into deeper clouds.

Our aim is to identify the height with the strongest particle growth induced by the strongest updraft. Herein, we have to avoid the influence of the entrainment at cloud top and that of precipitating ice towards cloud bottom. The MCAO clouds develop highly dynamically, and therefore, their depth spans a wide range (10th percentile = 30 m, 90th percentile = 625 m). Therefore, a fixed distance from the cloud top might not lie within thin clouds anymore. Thus, we consider a dynamic adaptation of the height.

We improved the text and moved the details into the Appendix. This includes a discussion about the height selection based on a sensitivity study (see Table A1).

Line 241-242: Does D constitute a cloud depth? Considering boundary-layer scaling, why not use the surface as a lower limit instead? Are the results sensitive to the -5 dBZ threshold to distinguish cloud from precipitation? What about when signals reach into the surface clutter.

D is the hydrometeor depth that includes cloud and precipitation hydrometeors. Taking the surface as a lower limit could result in the situation that the radar reflectivity is evaluated at a height where no cloud is present. The results are not sensitive to the -5 dBZ threshold as shown in Fig. A1. For all profiles, only signals above the blind zone (150 m) are evaluated. The same holds for the hydrometor depth that starts at 150 m regardless of whether signals reach below 150 m height or not.

Line 255: This is not a retrieval; "Extract" is better. Done.

Line 259 & 261: Number of samples are usually integers; there is no such thing as 2.9 samples. The fitting procedure provides the width in terms of the number of time steps (samples), which does not necessarily have to be an integer. For clarity, we mention the corresponding horizontal distance of 230 m in line 510.

Line 281: "particle concentration" is better; in reality I guess it has more to do with the cross-section area…We use particle concentration.

Line 285-287: All these precipitation or size against reflectivity relationships are a bit arbitrary; an expert on radar meteorology once told me that estimating ice concentration from radar reflectivity is uncertain to about an order of magnitude. So maybe phrase all this in a less distinct way.

The reviewer is completely righ. The dependency of the precipiation amount at Ny-Alesund on the Z-S relation was shown by Maahn et al, 2014. We emphasize the uncertainty stemming from the Z-S relation in line 151: „*Note that these S estimates are inaccurate since Z-S relations highly depend on ice habits, which are very variable within cloud streets (Maherndl et al., 2023a; Moser et al., 2023).*".

Line 296-306: From where is the information in this paragraph about ascending and descending air coming? Much of this appears speculative to me; besides drop "mass" - "air mass is a different thing.

The information about subsidence is from ERA5 reanalysis and now shown in Fig. 3g, h. We dropped the word "mass".

Line 309-311: This is a lot of words to say its colder on one day than the other; the temperature at 2 km has little impact on the boundary layer.

We highlight the exact temperature ranges because temperatures lie within and outside of the dendritic-growth zone on 01 and 04 April, respectively. The different temperature ranges might thus affect microphysics. Moreover, we mention the temperature above 2 km height to highlight that the temperatures are generally lower on 01 April and not only within the ABL due to processes within the ABL. The sentences in the updated manuscript are as follows: „*On 01*

*April, temperatures are lower than -20°C throughout all altitudes over sea ice (fetch <15 km)
and for parts over open water (Fig. 4a)*" (line 279) and „*all temperatures below 2 km height lie
within −20 to −10°C (Fig. 4f) and θ of the free troposphere is on average higher by about 5 K
compared to 01 April (Fig. 4b, g)*." (line 292).

Line 321: What do you mean by "weakens less"? Compared to what? No other weakening
inversion is mentioned in the text.

Rewritten: „ *the capping inversion over the sea ice close to its edge is stronger and weakens
less with fetch on 04 than on 01 April due to a layer of warm air above BLH.*" (line 295).

Figure 5. In fact, my interpretation of the temperature and moisture profiles is that the last profile
for 4 April shows a deeper boundary layer, at almost 1 km albeit less distinct, than on 1 April
that has a slightly more distinct top at maybe 600 m.

We do not share this opinion. The median of the cloud top height, which is often capped by
BLH, is 300 m and not 1 km. Due to wind shear, the air above this BLH came from Svalbard
while the air below originated from the north. Therefore, we strongly believe that air with
different characteristics was advected above ABL.

Line 327: "directional shear", "northerly" & "westerly". Done.

Line 328: I disagree; I can't see that the shear is systematically stronger for any of these heights

We have rewritten the sentence: „ A directional shear from northerly wind at the surface to
westerly wind occurs at all heights, which is strongest at BLH (Fig. 4i)." (line 295). The green
and blue lines in Fig. 4i show a change in wind direction at 150 and 400 m, respectively.

Line 329-330: Again, very many words to say there is a low-level jet at 200 m.

Rewritten: „*which is capped by a low level jet*" (line 282) and „*Even though a low level jet exists
at 200 m over sea ice as before (Fig. 4j), flow conditions differ compared to 01 April*" (line 294).

Line 332: I can't see any Ekman spiral here. Dropped.

Line 334-335: Explain the discussion on the different angles and the convective instability.

We dropped the discussion about the alignment of the cloud streets as it does not fit the
characterization of the ABL conditions with fetch.

Line 335-336: How does this statement agree with the previous angles discussed?

We dropped the discussion about the alignment of the cloud streets.

Line 341: It is not the profiles that precipitate!

Rewritten: "*On 04 April, Ze rarely exceeds -5 dBZ even below 500 m reducing the frequency of
precipitation compared to 01 April*" (line 308).

Line 351: "weaker advection" is better Done.

Figure 7: How is the normalized width calculated; mirroring in the center? See my question
earlier about the center (or the maximum updraft) appearing to be closer to one edge than the
other. Also, why no LWP for 4 April; there are LWP values in the fetch plot.

See answer regarding Fig. 2 on page 3 of this document.

We now show LWP in the Fig. 7 and 8.

Line 392-393: I suggest "… a factor of two or more …" Done.

Line 393: I have a hard time seeing any stagnation in this plot

Fig. 8 was revised and we hope that the new style simplifies seeing the reduced growth rate at fetches between 75 and 120 km.

Line 427-433: Suggest to drop this as you show no observations beyond 160 km.

Sorry, we meant 140 km which is included in Fig. 8. We corrected the statement in line 484.

Line 434-454: This can be dropped; if you decide to keep it shorten it and move to discussion or conclusions; it has no place here. We dropped it.

Section 5. I don't really see the value of this section here. Reading this after the enormously detailed previous sections, this feels incredibly thin.

Riming preconditions cloud and precipitation evoluion. Thus, we think it is an important aspect to study. We moved the analysis of riming to Sect. 4.2 as riming preconditions the conditions similar to the ABL conditions in Sect. 4.1.

Section 6.: The whole Synthesis can be shortened significantly

We deleted this section and shifted the most relevant information to other sections.

Line 534-550: Is this what you learned from this study; it seems exceedingly trivial and I could have told you all of this, saving you a lot of time. There has to be more than this!

We significantly reworked the introduction, including motivation and reformulation of the research questions. In this way also our conclusions were adapted to highlight the significance of our work.

Liu, A.Q., Moore, G.W.K., Tsuboki, K. et al. The Effect of the Sea-ice Zone on the Development of Boundary-layer Roll Clouds During Cold Air Outbreaks.Boundary-Layer Meteorol 118, 557–581 (2006). https://doi.org/10.1007/s10546-005-6434-4.

**Reviewer 2**

Thank you very much for spending the time and effort to thoroughly review our study, which helped to improve the manuscript. Based on the reviewer's comments, we rewrote most of the manuscript and shortened it significantly. We refined the research questions and give more insights into how our measurements are suited for model evaluation. Specifically, we worked on the description of the synoptic conditions during the MCAO event and of how the two flights probed different flavors of it. In the following, we reply (blue) to all reviewer's comments (black). Text passages from the manuscript are in italics. In our answers, we always refer to the line numbers of the revised manuscript.

One change that does not correspond to a reviewer's comment needs to be highlighted first. We found a problem in the roll circulation detection algorithm. For radar reflectivity (Ze) profiles with more than one cloud layer, the hydrometeor depth of the upper cloud layer was used by the algorithm previously. However, a few radar profiles observe two cloud layers with a very short cloud-free distance in between the layers, which likely occurs due to limited radar sensitivity. For these cases, the methodology to derive a roll circulation "object" ignored the lower cloud part. Thus, the updated version of the algorithm counts two cloud layers as one as long as the cloud-free distance between them is thinner than 50 m (see line 147). This update increases the hydrometeor depth for these cases. However, this modification of the algorithm only leads to small changes. Previously, we identified 364 and 109 cloud circulation objects in the 'cloud street' regime on 01 and 04 April, respectively. This changes to 356 and 112 objects which is corrected in line 251 .

**Major comments**

Abstract: Why these 2 CAOs in particular? Are they representative? Where do they fall within the more comprehensive assessment of Murray-Watson for example? This is addressed later in the document, but would be good to bring into the abstract. Is there a new process insight or hypothesis developed from their study that is new to their understanding? It is not clear to me why modelers should choose these examples above others, other than the opportunistic sampling. The authors also don't come back to this argument towards the end of the document.

We take these two cases because of two reasons: first, P5 had a special sampling strategy on these days flying perpendicular to the cloud streets. Second, even though both flights probed the same MCAO event (Walbröl et al. (2024)), both cases have different MCAO strengths that allow us to study the preconditioning by the strength of the MCAO.

We added to the abstract:

"*The evolution and structure were assessed by flight legs crossing the Fram Strait multiple times, sampling perpendicular to the cloud streets.*" (line 10).

 "*The two events, just three days apart, belong to a particularly long-lasting MCAO and occurred under relatively similar thermodynamic conditions. However, for the first event, colder airmasses from the central Arctic led to an MCAO index twice as high as for the second event*" (line 6).

The setting with a high over Greenland and a low over Siberia is characteristic for most MCAO over the Fram Strait as shown by Dahlke et al (2022). Nevertheless, both cases

are stronger than the 75th percentile of the MCAO index between 1979 and 2022 (Walbröl et al. (2024)). We also added information on the synoptic situation in Sect. 2.1.

To the abtract, we added: "*though both events were stronger than the climatological 75th percentile for that period.*" (line 8).

We added more discussion on the study by Murray-Watson et al. (2023) who analysed MCAO development over 30 hours in the introduction. We also added a time axis to Fig. 8, which highlights that we focus on the first four hours of the events. Contrarily to their study, for our cases mixed-phase clouds dominate (Tab. 2).

The revised text includes more details on why modelers should choose our cases. With our data, modelers can test whether their model can represent dynamics and microphysics at the same time. Moreover, our data offer the unique opportunity to test the MCAO evoultion of models close to the sea ice edge.

We have added these information to the abstract and made the argument about the model comparison study at the end of the manuscript: „*Within our analysis, we developed statistical descriptions of various parameters (i) within the roll circulation and (ii) as a function of distance over open water. In particular, these detailed cloud metrics are well suited for the evaluation of cloud-resolving models close to the sea ice edge to evaluate their representation of dynamics and microphysics.*" (line 21)

„*To answer the two last research questions, we established composite approaches to characterize the roll circulation (Fig. 7) and fetch (Fig. 8) depends. Such metrics can also be generated from cloud-resolving model output and be used to evaluate their performance to represent microphysics and dynamics in the initial phase of an MCAO. By considering the two cases with similar large-scale synoptic settings but differences with respect to microphysics, e.g., LWP and riming, insights into the simulation of cloud microphysics could be gained. In particular, it will be interesting to analyze whether such models successfully reproduce the observed factor of two in scaling found for several parameters between the two cases.*" (line 463)

It makes sense to me that the 2nd CAO is the weaker of the 2, as it occurred 3 days later, and likely experienced similar synoptic conditions. The authors treat the 2 CAOs as if they are independent of each other. What initial conditions did they share? A figure indicating the dominant synoptic situation and how it evolved would helpfully complement the first figure. I hypothesize a coastal low interacted with the northerly CAO flow to develop that convergence line, for example.

Both days were part of a longer MCAO which lasted for more than two weeks (see Walbröl et al., 2024 and their Fig. 11). During this period, slight changes in MCAO intensity occurred, though both days were above the 75[th] percentile. We now show the synoptic situation on both days in Fig 1a, b and discuss the environmental conditions in Section 2.1. The major difference is that on 04 April, the flow came more from the east and was thus affected by Svalbard.

**More detailed comments:**

Abstract, line 12: "the liquid layer is wider" is unclear here. Do you mean a larger horizontal dimension of the rolls? Except the previous sentence says the width of the roll circulation is similar between the 2 cases. "Occurrence" -> occurrence of what? "are" -> "have"

We have dropped the first statement and rewritten the other ones, i.e., precipitation occurrence (line 15).

Line 30: "them" what is 'them'? Define

The statement was removed when we shortened the introduction.

Lines 33-36: unclear to me how the 2nd sentence relates to the first. Isn't the first sentence saying vapor depositional growth is favored when the vapor pressure> ice and liquid saturation?

Sentence removed during revision.

Line 37: awkwardly written. The statement was removed when we shortened the introduction.

Line 73: "were" -> "was" Done.

Line 83: "extend" -> "extent". Same for line 391. And line 520. Done.

Line 85: awkward. Overall the sentence is vague. What are the induced changes in dynamics and clouds?

The statement was removed when we shortened the introduction.

Line 160-161: how do you know most of the liquid is near the top? As written this sentence sounds presumptuous. And what is the motivation from deriving LWP from the slanted profiles? Superficially the nadir profiles would make more sense.

We rearranged the data section to describe the retrieval of the lidar cloud top height, which is sensitive to the liquid on top, before the radar cloud top height. A discussion on the typical structure of mixed-phase clouds can be found in Shupe et al (2008).

We state: "*Lidar backscatter is highly sensitive to hydrometeors, especially to liquid, which, in our case, is always super-cooled.*"(line 123)

"*Cloud top height is also derived from the radar profiles at the height of the uppermost radar reflectivity signal above the noise level. Comparing this height with CTH from lidar allows us to assess the supercooled liquid layer thickness (LLT). Here, we exploit the fact that the lidar is more sensitive to particle amount (liquid), whereas the radar is more sensitive to particle size, i.e., ice particles (Ruiz-Donoso et al., 2020).*" (line 141)

"*As shown by the lidar backscatter and its strong attenuation close to cloud top and in accordance with Shupe et al. (2008) we assume that most liquid resides in the uppermost few hundred meters of the cloud.* " (line 158)

You are absolutely right, nadir profiles would be the best. However, the LWP is only available on slanted paths and contrarily to the radar profiles not corrected to nadir. Thus, we correct the profiles to nadir via geometrical considerations.

In the manuscript we state:

"*Both measurements are taken with 25° backward inclination of the instruments. While the vertically resolved radar measurements are reconstructed to nadir measurements, the passive measurements represent a slant path.*" ( line 130)

"*While radar reflectivities are corrected to nadir profiles, the TB and thus LWP measurements are along the slant path (Mech et al., 2022a).*" (line 157)

Line 240: wouldn't the strongest updrafts generate liquid drops that are less obvious to the radar than the radiometer? Why expect ice production and liquid production to be collocated?

We expect ice production within updrafts because there saturation with respect to ice frequently occurs *(Korolev and Field, 2008)*.

See line 234: "*Here, frequent saturation with respect to ice and thus the formation of cloud droplets and growth of both liquid and ice particles occurs (Korolev and Field, 2008).*"

Line 256: 2.9 samples = what horizontal distance?
This corresponds to a horizontal distance of 230 m (line 510).

Fig. 4: this is the first introduction of the measurement regimes, so you may want to identify them in the caption, it's hard to understand this figure otherwise. Are these the same as provided in Table 1? the legend and symbol shapes are hard to see. Besides increasing them in size, would also suggest increasing the size of the overall figure.
Overall, the new layout of the figure does not include the regimes anymore.

Line 281: what is the uncertainty in the LLT metric?
The uncertainty in LLT is difficult to quantify. Geometrically, the vertical resolution of the instruments (e.g., 7.5 m for the lidar) limits the lower end.  See manuscript line 144: „*Due to limited vertical resolutions of the instruments and resulting uncertainties in CTH, the CTH of the lidar has to exceed the CTH of the radar by at least 10 m to be defined as liquid topped*".

Line 290: "precipitating" -> precipitation Done.

Section 4.1: there really should be a synoptic analysis and figure provided, this would be a good spot for it. Done, see Sect. 2.1 and Fig. 1a, b.

Line 296: was VIIRS imagery, of higher spatial resolution, not available?
We show the corrected reflectance in true color of Modis that uses bands 1, 3, and 4, which have a resolution of at least 500 m. The bands (l1, m4, m3) that are used for the true color corrected reflectance obtained by VIIRS have a resolution of 375 and 750 m. Thus, this product is not finer resolved than the respective Modis image having a resolution of at least 500 m. The corrected reflectance of VIIRS in false color (that is shown in Fig. 1g, h) takes the bands m3, l3, and m11 into account. Here, the m bands have a resolution of 750m, and the l bands of 375m. The corrected reflectance from Modis is thus finer resolved.

Table 2 is nice. Does the BLH top correspond to the cloud top, or the mixed-layer height? Indicating the range of cloud top heights would be useful also.

We now provide the acronyms in the Table caption and added as further information the median and the interquartile range (IQR) of the cloud top height. "*BLH and CTH stand for atmospheric boundary layer height, i.e., the inversion height of potential temperature, and cloud top height, respectively.*". Boundary layer height and CTH are thus derived independently from dropsondes and radar, respectively.

Lines 301-304: fig. 4 is difficult to follow to be honest. The dropsonde and ERA5 values don't agree well, the disagreement is more than I would have expected based on Seethala et al. 2021.

We updated the old Fig. 4 by the new Fig. 3 which helps to compare dropsonde and ERA5 data for MCAO indices and sensible heat fluxes. Generally, we can see a good spatial consistency of both.

Line 308: I don't see mean thermodynamic profiles provided as part of fig. 1e,f,.....these figures only indicate the locations of the dropsondes.
Sorry, the reference did not fit here. We dropped the reference.

Line 303: One also can't see from Fig. 4 that the MCAO follows the SST (SST isn't shown) and basically we are just seeing a few values for the ERA5 dataset, because of the coarse resolution. I would suggest increasing the size of the dropsonde-derived values in fig. 4, and including SST, and reducing the size of the ERA5 values. I am confused why there are so many ERA5 values. The interesting aspect of this figure is the sharp increase in the MCAO index for 1 April (isn't it)? How much of the fetch is over the MIZ? It would be useful to include the beginning point in the figure 4.
The layout of Fig. 4 (now Fig. 3) has completely changed. The updated panels include also SST and the locations with 80 % sea ice concentration. Still, the differences between MCAO indices retrieved from dropsonde observations and ER5 data are visible close to sea ice edge that arise due to the less sharp increase in ERA5.

Line 314: where are you showing the surface flux values you refer to?
We integrated ERA5 fluxes into Fig. 3 and also calculated them from dropsondes. The reference has been adapted.

Line 315: here you define the BLH. This would be useful to include in the caption of table 2 as well.
Done.

Lines 328-330: I rather think a spatial plot of the ERA5 winds and Geopotential height would be more informative here.
We now provide a map of the geopotential potential height at 500 hPa, mean sea level pressure and 850 hPa equivalent potential temperature in Sect. 2.1 (Fig 1a, b). This plot does not include wind fields as they can be derived from the shown variables.

Line 339: that the clouds only reach 1km and 500m respectively is interesting too. The CAOs documented at COMBLE tend to reach higher.

Yes, we now discuss this in more detail in our introduction. During COMBLE, observations were taken at around 500 and 1000 km fetch while we are focusing on the first 150 km. Thus, they capture a later stage of evolution than our data. This might explain the higher cloud top heights during COMBLE. For one MCAO during COMBLE, Geerts et al. (2022) retrieved from satellite observations that the BLH close to sea ice edge reaches also only a few hundred meters.

See line 65: "*Therefore, the Cold-Air Outbreaks in the Marine Boundary Layer Experiment (COMBLE) in 2021/2022 (Geerts et al., 2022) established two ground stations at Andenes and Bear Island, Norway, which provided important insights into cloud properties (Mages et al., 2022; Lackner et al., 2023) and supported model evaluation (Geerts et al., 2022). However, these stations were located about 1000 km away from the sea ice edge. Thus, only open and closed cellular convection but no cloud streets have been observed.*"

Line 342: awkward writing.

Rewritten: „*The shorter the fetch on 01 April, the stronger is the decrease in the mean Ze profile close to the surface (not shown).*" (line 304).

Line 343: you could look at the near-surface RH to examine if more evaporation is happening.

We investigated the profile of the relative humidity with respect to ice (see Fig. R1 at the end of this document). The figure nicely shows that for small fetches on 01 April (a, light blue), the relative humidity decreases to values below 100 % close to the surface. This confirms the hypothesis that ice particles might sublimate under these conditions.

In line 305, we added a statement to the manuscript: „*Thus, near-surface ice particles might experience stronger sublimation on 01 April when the mixing ratio is comparably small and relative humidity with respect to ice below 100 % (not shown).*" .

Line 347: not convinced this is the best place to mention Morrison et al 2012. That is a modeling study based on more weakly-forced clouds. If you are going to mention it, better left to a discussion later on. We dropped the sentence.

Fig. 7: this is a nice figure. The caption should state clearly what the 3 groupings represent. Is it the same for all panels? The lowest one is labeled differently. How wide/coarsely resolved are these distances? I just wonder if ice production is truly colocated w liquid production, or perhaps they get lumped together through a coarseness in the grouping.

We agree that it was irritating that the x-axis was not the same for all panels. The axis of the first row differed from the other rows. Thus, we separated the first row from the rest of the figure and put it into the supplement.

In the caption, we state: „*The median (horizontal line) and lower and upper quartile (box edges) are displayed at the boundary of the clouds, the updraft position, and in between.*"

Moreover, we added to line 349: "*More precisely, we group cloud properties according to their distance from the maximum updraft region ($Ze_{0.7}$) into three regions: the central updraft region, the region close to cloud boundary, and the region in between.*"

The resolution of the bins depends on the width of the cloud streets and is not universal. It might be true, that ice and liquid production is lumped together. However, since we take cloud streets that are at least 5 radar profiles wide, a finer grouping is impossible.

Line 367: not following this sentence. What do you mean by extreme?

Rewritten: „*Strong riming events might explain the frequent high extremes of S (snowfall).*" (line 360).

Line 368: 87 -> 87% Done.

Line 370: spell out what '0.6 of D' means (it's easy to lose track of the acronym meanings).

Done.

Line 375: is the measurement limit known to the reader somewhere earlier? I don't recall seeing it.

The measurement limit is already known to the reader, it was first introduced in Sec. 2.2, line 156: "*Depending on atmospheric conditions, the maximum uncertainty is below 30 $gm^2$ (Ruiz-Donoso et al., 2020).* ".

Line 380: I think the clouds looked at within Shupe 2008 are different enough from CAOs that this analogy isn't quite right. It could just be a coincidence. You could say in the discussion section that examining if CAO clouds are a truly unique cloud regime that behave differently from non-CAO MPC is a topic worthy of future study (I suspect the differences in surface fluxes do genuinely separate CAO and non-CAO clouds).

We dropped this statement as an analogy cannot be drawn.

Lines 381-383: if you want to keep this, it would be better suited for a discussion section towards the end.

We dropped this paragraph.

Line 398-399: vague. Might help to indicate the distances of 'large' and 'small' fetches.

We changed to: „*The aspect ratio of the circulation decreases with fetch for fetches smaller 50 km and stays constant for larger fetches.*" (line 459).

Line 407: to me it makes more sense to look at how cloud microphysics depends on the surface fluxes, as opposed to the exposure to open water. You can actually estimate the surface fluxes from the dropsondes, this was done within Seethala et al 2021. It would be interesting to know those values.

Following Seethala et al. (2021) we now calculated the fluxes from dropsondes and compared them to ERA5 data in Fig. 3. A discussion can be found in line 259 ff. The figure shows the strong changes in flux strength close to the sea ice edge and a good consistency between dropsondes and ERA5. As the dropsonde locations were rather different on both days, ERA5 is better suited to get the overall picture.

Line 408-409: this sentence doesn't make sense.

Rewritten: „*On this day, 90 % of the profiles containing liquid-topped cloud streets have LLT smaller than 100 m, which is more than on 01 April (70 %).*" (line 409).

Line 413: here is where I think the reader learns the LWP uncertainty. Better to say the LWP values are beneath the maximum uncerinty of 30 g/m2.

Here, we removed the statement. However, the measurement limit is already known to the reader, it was first introduced in Sec. 2.2, line 156: "*Depending on atmospheric conditions, the maximum uncertainty is below 30 gm$^2$ (Ruiz-Donoso et al., 2020).* ".

Lines 427: how do you deduce the subsidence? From satellite imagery?

We deduce subsidence from ERA5 and show it in Fig. 3. In the manuscript we write: „*Even though ERA5 reanalysis with its coarse resolution shows a rising air mass at these fetches (Fig. 8), we suggest that the air mass subsides and suppresses cloud development.*" (line 385)

Lines 434- : I appreciate the effort to place the 2 CAOs into context and not claim the noted features are universal (though they well might be). It would be nice to see this sentiment expressed in the abstract as well.

See our answer to your first major point

Line 445: decreaseing -> decreasing Done.

Line 452: Khanal et al 2020 examine MPCs using MODIS, cited below.

We dropped the statement. We do not cite Khanal et al. (2020), because their analysis did not focus on Arctic MPCs.

Line 475: since updraft speeds are correlated to surface fluxes, you could examine the dependence of riming on dropsonde-derived surface fluxes to put this statement on a stronger footing. Unless you have in-situ vertical velocities from one of the planes (it doesn't seem like it though).

The variability in riming is on a spatial range of 700-1100 m while the dropsondes probe at distant locations compared to the aircraft with sondes falling for roughly 5 min through the atmosphere. Thus, matching radar measurements and sondes is rather complex.

Line 485: the -> these, before 'conditions' Done.

Lien 486: do you show CTH<BLH somewhere? I don't recall seeing that.

In line 377 we state: "*The comparison of boundary layer height BLH derived from dropsondes and closely located airborne measurements showed that CT H is generally only by 8.5 m lower than BLH*"

Line 508: isn't dendritic growth through vapor deposition, followed by aggregation, a different growth process than riming? I'm not sure I understand how aggregation encourages more riming.

Sorry, some poor English phrasing led to a misunderstanding. We reframed the sentence as follows: „*On this day, cloud top temperatures are colder than or at the low*

*end of temperatures within the dendritic-growth zone (DGZ; −20 to −10°C) and hence too cold for aggregation to be dominant (Chellini et al., 2022). On 04 April, contarily, riming is not significant because cloud top temperatures lie within the DGZ that favors aggregation.*" (line 329).

Line 508: and how do you know SST is rising? You haven't shown SST anywhere.

We included SST maps from ERA5 in Fig. 3a, b.

Line 510: this is difficult to follow, not sure what this sentence is saying.

We have simplified the sentence: „*In contrast, a decrease of LLT (32 %) by 20 m can be seen.*" (line 355).

Line 511: I think liquid droplets are typically more readily raised, because they are smaller than ice particles. Or maybe you just need to be more precise about the particle size - but I'm not sure you have the microphysical measurements to back up this statement. All in all, this paragraph is too speculative. Please dial it back.

We now emphasize that the statement is just our hypothesis: „*We hypothesize that updrafts transport ice particles into higher parts of the clouds. The mixed-phase region thus increases at the expense of the liquid layer and riming is enhanced (Fig. 4.2). Riming increases ice particle size, Ze, and S in updrafts. The observed LW P increase in updrafts might indicate that condensation is more favored than depletion of liquid.*" (line 356).

Line 516: you are contradicting what you said on line 508.

This is again due to the misunderstanding. As already mentioned, we have rewritten the sentence.

"*On 04 April, contrarily, riming is not significant because cloud top temperatures lie within the DGZ that favors aggregation.*" (line 331).

Line 549: I don't believe temperatures have to be colder than -20C for riming to be active, this is certainly not universally true, nor do you explicitly show this.

You are right, we do not explixitly show it. We deleted the statement.

[Figure]

Fig. R1: Averaged dropsonde profiles from HALO and P5 of relative humidity with respect to ice ($RH_{ice}$) binned by fetch on 01 April (a) and 04 April 2022 (b). The shaded areas represent the standard deviation of each category. The color coding follows the categorization of the manuscript.

---

## Author Response (AR2)

**Reviewer 2**

Thank you very much for taking the time and investing the effort to thoroughly review our study. The raised comments helped to further improve the manuscript. In the following, we reply (blue) to all reviewers' comments (black). Text passages from the manuscript are in italics. In our answers, we always refer to the line numbers of the newly revised manuscript.

line 39: I don't think Morrison et al 2012 were exploring mixed-phase cloud streets. In their study the super-cooled liquid clouds were stratiform and long-lasting because ice depletion was low. The Murray-Watson paper would be a more topical reference here.

Due to retrieval limitations, Murray-Watson et al. (2023) only considered liquid-dominated clouds as well (see line 59 of the manuscript). Thus, we added Abel et al. (2017) as a reference, who studied airborne precipitation observations of mixed-phase clouds during a MCAO event.

Line 229: why not also correct the passive LWPs for the slant path angle and provide a vertical LWP estimate? It's true the LWP values may not be exactly accurate, but the same can be said for the geometrically-corrected radar measurements. It is difficult for the reader to hold on to the idea that the reported LWP values are all along a slant path rather than vertical. This process of correcting radar but not passive microwave for the slant angle doesn't make sense to me. Later on you discuss shifting the LWP values in time to bring them into closer agreement with the radar. To my mind simply correcting both the radar and microwave similarly for the same geometrical issue makes more sense.

We thank the reviewer for bringing up this important point which made us revisit our retrieval theory described in detail in Appendix B of Ruiz-Donoso et al. (2020).
 A slant geometry is necessary for the MiRAC FMCW radar to avoid strong surface returns from the active component. As described in detail in Mech et al. (2019), the radar profiles are reconstructed to nadir.
 For the passive component, meanwhile, it is less straight forward to correct for the slanted geometry. The measured brightness temperatures correspond to the sum of signals originating from the entire slanted atmospheric column within in the viewing cone of the instrument, including sources from surface, atmosphere, and clouds/precipitation.
 Correcting the brightness temperatures to nadir would require making assumptions on the (unknown) vertical setup of the atmosphere and the ground, as well as running radiative transfer calculations to forward simulate the measured signal. We believe that this would induce even higher uncertainties compared to the present approach of shifting the retrieved LWP in time. The vertically integrated liquid water observed by MiRAC-A on slant geometry is instead derived using the following method.
We set up an atmosphere based on the dropsondes of the campaigns with many artificial cloud profiles. Based on this large set of atmospheres, we perform radiative transfer simulations to get a solid database for liquid water paths and corresponding brightness temperatures for various slant geometries, and derive cubic regression coefficients. To derive the liquid water path from the brightness temperature observations, we apply the coefficients from angles and atmospheric altitudes closest to the aircraft observations of each brightness temperature. This method is further described in Appendix B of Ruiz-Donoso et al. (2020). Moreover, a manuscript on the LWP retrieval is in preparation.

We additionally tested the validity of the applied spatial shift of the LWP retrieval obtained from geometrical considerations. We therefore compare the brightness temperatures obtained by MiRAC-A (89 GHz) with brightness temperatures from HATPRO (31.4 GHz as  89 GHz is not available),

another radiometer that is installed onboard the P5 in nadir-looking geometry. Comparisons show a good accordance between the shifted and nadir measurements for both days as shown in Fig. R1 for a short time series on 04 April.

[Figure]

*Figure R1: Time series of the brightness temperature on 04 April obtained by MiRAC at 89 GHz and HATPRO at 31.3 GHz. Left: originally obtained data that are inclined by 25° for MiRAC and nadir for HATPRO. Right: Original HATPRO data (nadir) and spatially shifted MiRAC data that mimic the nadir view.*

Note that we do not take the HATPRO observations for deriving LWP here in order to be able to compare the retrieved LWP with consistent estimates from past and future campaigns within the AC³ framework which always included MiRAC measurements.

Line 258-260: incomplete sentence here beginning with 'While'.
We corrected the sentence: „*While radar reflectivities are corrected to nadir profiles, TB and, thus, LWP measurements are measured along a slanted path (Mech et al., 2022a).*" (line 158)

Line 308: it would be worth assessing if ERA5 fluxes (which you also use, e.g line 434) match those calculated as you did from the dropsondes, building on Seethala et al 2021. The accuracy of ERA5 turbulent fluxes over the open Arctic Ocean is not well known, and this is an opportunity to opine on how well they follow those calculated from the dropsondes using the COARE 3.5 flux algorithm.
We thank the reviewer for raising this important comment. We point the reviewer to Fig. 3 e, f and line 270: „In accordance with Seethala et al. *(2021), fluxes and MCAO indices from ERA5 generally correspond to dropsonde estimates, except over sea ice where ERA5 seems to overestimate the fluxes. Finer spatial structures in both parameters are resolved in the dropsondes.*"

Line 545: how are the in-situ particle shape measurements determined? Is this a fractal dimension?
The optical in-situ particle shape measurements were obtained by the instruments CDP, CIP, and PIP. More information on the instruments and derived properties is given in lines 166 ff. A fractal shape of the particles is derived. The full complexity of the retrieval is described in Maherndl et al (2024).
For clarity, we added the following to line 170: "*Rimed mass is calculated from images of the fractal particle shapes, as well as the continuous particle size distribution derived from combining*

*CDP, CIP, and PIP observations."*

Line 561: where is this suggestion of more riming -> more precip supported? It seems reasonable other than that faster-falling particles might be able to reach the surface without a phase change.
We agree with the reviewer and removed the corresponding statement from the manuscript. We additionally adapted the text which now reads (line 340): „*For 01 April, we, hence, infer that riming is mainly present within the updraft regions of cloud streets. A more detailed comparison with λ of the roll circulation detected by the remote sensing measurements is performed in Sect. 4.4*"

Line 650: remove 'by'
Done.

Line 844: should 'III' be 'II'? This is the 2nd question right? Revisit subsequent numbering also if so.
Done.

Line 852: assuming the data support the hypothesis, please try to more clearly connect the hypothesis to the supporting data.
The updated manuscript states: „*Our statistical analysis of median cloud characteristics within the roll circulation and their variability (Fig. 7) could be used to test the performance of cloud parameterizations and better understand riming effects.*" (line 466).

Refs:
- the Maherndl et al 2023 now has a final revised paper that would be better to cite than the preprint.
Done.
- Seethala, C. Et al, 2021: On assessing ERA5 and MERRA2 representations of cold-air outbreaks across the Gulf Stream. Geophys. Res. Lett., 48, doi:10.1029/2021GL094364
Done.

Ruiz-Donoso, E., Ehrlich, A., Schäfer, M., Jäkel, E., Schemann, V., Crewell, S., Mech, M., Kulla, B. S., Kliesch, L.-L., Neuber, R., and Wendisch, M.: Small-scale structure of thermodynamic phase in Arctic mixed-phase clouds observed by airborne remote sensing during a cold air outbreak and a warm air advection event, Atmos. Chem. Phys., 20, 5487–5511, https://doi.org/10.5194/acp-20-5487-2020, 2020.

Maherndl, N., Moser, M., Lucke, J., Mech, M., Risse, N., Schirmacher, I., and Maahn, M.: Quantifying riming from airborne data during the HALO-(AC)[3] campaign, Atmos. Meas. Tech., 17, 1475–1495, https://doi.org/10.5194/amt-17-1475-2024, 2024.

**Reviewer 3**

Thank you very much for taking the time and investing the effort to thoroughly review our study. The raised comments helped to further improve the manuscript. In the following, we reply (blue) to all reviewers' comments (black). Text passages from the manuscript are in italics. In our answers, we always refer to the line numbers of the newly revised manuscript.

Major Concern

Sections 4.3 and 4.4 contain a few portions where the authors discuss their findings, and it is difficult to separate actual findings from speculations. The authors should either use more nuanced language (especially in the portions immediately following a hypothesis) or prepare a separate discussion section:
- ll. 367-370 The authors first suggest ("seems to") the role of MCAO strength and then imply ("thus suppress") with further causal implications after.
- ll. 379-381 This sentence contains mere speculation.
- ll. 356-359 After suggesting a relationship ("We hypothesize"), the authors imply certain properties ("thus increases")
- ll. 408-415 This paragraph is filled with discussion elements.

We thank the reviewer for this comment and thoroughly went through the manuscript, therein especially Sec 4.3 and 4.4, to adjust the language as needed. In more detail, we updated the following sentences, and revised the corresponding sections to account for language changes.

*„The smaller MCAO strength on 04 April seems to weaken the updraft motion and might, thus, suppress the rise of CTH and the lifting of ice into the liquid layer in updrafts. In updrafts, this might prevent riming, likely hampering an increase in S and mean Ze as well as a lifting of the height level with highest ice occurrence."* (line 371).

*„A potential reason for this reduction might be a reduced buoyancy in the ABL, and warm air being advected above the boundary layer. Future modeling experiments could test this hypothesis, including lee effects on air mass development caused by the Svalbard archipelago. "* (line 385).

*„We speculate that, here, updrafts carry ice particles to higher cloud regions. If so, the mixed-phase region would expand at the expense of the liquid layer and would enhance riming (Fig. 4.2). Potential riming occurrence would increase ice particle size, Ze, and S in updrafts. The observed slight LWP increase in updrafts (Fig. 7e, f) could indicate that, in our study, condensation is more favored than depletion of liquid."* (line 359).

*„While the evolution of cloud microphysics with fetch is similar on both days, thermodynamic conditions modify the intensity of the parameters.*
*On 04 April, characterized by overall warmer temperatures, clouds are more shallow. On this day, 90% of the profiles containing liquid-topped cloud streets have LLT of smaller than 100m, which is more than on 01 April (70%). Less supercooled liquid may reduce the amount of liquid-topped cloud profiles (Table 2), LWP (Fig. 8k) and LLT (Fig. 8i). A potential mechanism could be that the warmer temperature, low amount of supercooled liquid, and weak MCAO index prevent riming, reducing  snowfall rate and mean Ze. This could potentially explain why snowfall occurs less frequently on 04 April. Moreover, the lack of riming in updrafts would reduce the variability in snowfall rate within each fetch bin. Lacking preconditioning by riming might delay the precipitation onset on 04 April by more than 10km (Fig. 8p)which starts forming at fetches of 26 and 39 km on 01 and 04 April, respectively. "* (line 419).

Minor Concerns

ll. 22-23 Thinking of quasi-Lagrangian simulations that could be evaluated, it would be helpful to provide information regarding the boundary layer windspeed as well as the spatial and temporal gap between subsequent crosswind flight legs. Are flight legs really revisiting an air mass or are stationarity assumption needed here?

*We thank the reviewer for the clarifying question. The impact of wind speed is discussed in Fig. 4, and flight legs are discussed in Sect. 2. The flight legs probe the same locations several times, they resemble each other, and have roughly the same length. No spatial gaps between adjacent flight legs exist. Hence, flight legs revisit the same location but not the same air mass.*
*In the abstract, we specified: „The evolution and structure were assessed by flight legs crossing Fram Strait multiple times at the same location, sampling perpendicularly to the cloud streets." (line 10).*

l. 25 Maybe remove "accompanied by" or I'm missing the point and more explanation is needed here.

*Done.*

ll. 54 I tend to disagree and would soften this statement. There have been earlier and simultaneous efforts that explore MCAOs using satellite data in a quasi-Lagrangian manner (Wu and Ovchinnikov, 2022, Tornow et al., 2023).

*We softened the statement: „While cloud reflectance measurements by satellites have provided important insights into the geometrical appearance of MCAOs since their beginning, recent studies such as Murray-Watson et al. (2023), Wu and Ovchinnikov (2022), and Tornow et al, 2023 quantitatively studied cloud development in a quasi-Lagrangian way." (line 52)*

l. 222 Please specify "undisturbed".

*We dropped undisturbed.*

l. 223 Please specify the panel(s) within Fig. 1 that the color can be found in.

*Done.*

ll. 282-283 Please specify where to find the low-level jet.

*We modified the sentence: „The ABL is capped by a low-level jet at 250m height (Fig. 4e)" (line 285).*

Fig.3 (and also Fig. 4): I highly recommend showing subsidence at a level that is aligned with the cloud-top (and its free tropospheric entrainment). By virtue of looking at the surface layer, subsidence is essentially zero and can have no practical meaning here.

*We thank the reviewer for this important remark and updated Figure 3. The updated version shows the subsidence at the median cloud top height of each day, i.e., 925 and 975 hPa on 01 and 04 April, respectively. Since we only retrieved the cloud top height along the P5 track and are not able to do so for all locations shown on the map, we decided to show subsidence at a constant pressure level for each day.*
*We rewrote parts of the analysis to clarify this (line 275):*
*„At the height of the median CTH, here 925 hPa, air subsides within the 'prior to cloud streets' and 'cloud streets' regime, respectively (Fig. 3g, green and blue track). Over ocean, subsidence is generally reduced compared to over sea ice. The area of fetches between 75 and 120km around 7°E longitude is characterized by strong subsidence (Fig. 3a, c) throughout the entire atmospheric column (not shown) despite increasing SST and MCAO indices. This wave-like pattern is likely induced by wave effects originating from Svalbard archipelago (Shestakova et al, 2022).*

*„Compared to 01 April, the air mass at CTH (975 hPa) ascends for fetches larger than 60 km (Fig. 3h). A wave effect is notable within the region affected by the lee effect but not for the analyzed data west of the convergence line."* (line 293).

Fig. 3g and h (and also Fig. 1 e and f): I'm confused as to why only part of the flight track is shown (that evidently looks more complex as for example displayed in Fig. 1g). Perhaps the authors should mention why they only show a subset.
The mentioned Figures show the complete flight track of the Polar5 aircraft. The blue line in Fig. 1g depicts the longer and more complex track of HALO instead.

Fig. 4: I recommend a different set of colors here as they are hard to tell apart.
For clarity, we use the same color coding as in the plots before. To not have two blue colors in the first row, we changed the color of the cloud street observations with large fetches to gray.

Fig. 7 and 8: I recommend adding cloud droplet number concentration that is expected to decrease where riming and precipitation are active.
Cloud droplet number concentration was only observed onboard the aircraft P6 and radar observations, from which we retrieve up- and downdraft information, are only availabe for P5. Since we do not know the up- and downdraft regions for P6 observations, the composit shown in Fig. 7 cannot be generated for cloud number concentration. Moreover, a closer focus on in situ observations would be beyond the scope of this study.

ll. 357-358 Could the reverse be true, too (that is, greater ice particle size increases riming)?
The reverse might be also true as long as the rimed particles do not grow too large and thus do not sediment away from the liquid layer, which we can not prove. We did not add this speculative statement.

ll. 361-362 Why is most ice expected at Zemax = 0.6? More information seems needed here.
We explain why we assume that most ice is located at the height of the maximum *Ze* per profile in Sect. 3.2. According to Fig. 7k, this height is at 0.6 of the hydrometeor depth for updraft positions on 01 April.

ll. 379-381 Could CTH also be affected by a change in subsidence (e.g., Tornow et al., 2023) ?
At the pressure level closest to cloud top height, air generally subsides on 01 April and ascends on 04 April, respectively, as illustrated in Fig 3g, h. This subsidence pattern does not explain why CTH are on average shallower on 04 April. Thus, we assume that the general differences in CTH between the days originate from other factors discussed in more detail in line 385. Local variations of CTH in different magnitudes of fetch and their dependence on subsidence are discussed in manuscript line 389.

ll. 382-384 Again, I recommend using subsidence at CTH altitude to obtain a meaningful assessment.
See answer above.

Typos

l. 22 Maybe change to "These detailed cloud metrics are particularly well suited…"
Done.

l. 229 Please change to "80% < SIC < 100 %".

Done.

There are punctuation errors throughout (e.g., l. 20, ll. 232-234, l. 148). I recommend asking a native English speaker for their input.
Done.

References

Tornow, F., A. S. Ackerman, A. M. Fridlind, G. Tselioudis, B. Cairns, D. Painemal, and G. Elsaesser (2023), On the Impact of a Dry Intrusion Driving Cloud-Regime Transitions in a Midlatitude Cold-Air Outbreak. J. Atmos. Sci., 80, 2881–2896, https://doi.org/10.1175/JAS-D-23-0040.1.

Wu, P., & Ovchinnikov, M. (2022). Cloud morphology evolution in Arctic cold-air outbreak: Two cases during COMBLE period. Journal of Geophysical Research: Atmospheres, 127, e2021JD035966. https://doi.org/10.1029/2021JD035966